# Personalised Federated Learning On Heterogeneous Feature Spaces

**Alain Rakotomamonjy**                                            *alain.rakoto@insa-rouen.fr*
*Criteo AI Lab*
*Paris, France*

**Maxime Vono**                                                   *m.vono@criteo.com*
*Criteo AI Lab*
*Paris, France*

**Hamlet Jesse Medina Ruiz**                                      *hj.medinaruiz@criteo.com*
*Criteo AI Lab*
*Paris, France*

**Liva Ralaivola**                                                *l.ralaivola@criteo.com*
*Criteo AI Lab*
*Paris, France*

**Reviewed on OpenReview:** *https://openreview.net/forum?id=uCZJaqJchs*

## Abstract

Personalised federated learning (FL) approaches assume that raw data of all clients are defined in a common space *i.e.* all clients store their data according to the same schema. For real-world applications, this assumption is restrictive as clients, having their own systems to collect and then store data, may use *heterogeneous* data representations. To bridge the gap between the assumption of a shared subspace and the more realistic situation of client-specific spaces, we propose a general framework coined `FLIC` that maps client's data onto a common feature space via local embedding functions, in a federated manner. Preservation of class information in the latent space is ensured by a distribution alignment with respect to a learned reference distribution. We provide the algorithmic details of `FLIC` as well as theoretical insights supporting the relevance of our methodology. We compare its performances against FL benchmarks involving heterogeneous input features spaces. Notably, we are the first to present a successful application of FL to Brain-Computer Interface signals acquired on a different number of sensors.

## 1 Introduction

Federated learning (FL) is a machine learning paradigm where models are trained from multiple isolated data sets owned by individual agents/clients, where raw data need not be transferred to a central server, nor even shared in any way (Kairouz et al., 2021). FL ensures data ownership, and structurally incorporates the principle of data exchange minimisation by only transmitting the required updates of the models being learned. Recently, FL works have focused on *personalised* FL to tackle statistical data heterogeneity and used local models to fit client-specific data (Tan et al., 2022; Jiang et al., 2019; Khodak et al., 2019; Hanzely & Richtárik, 2020). However, most existing personalised FL works assume that the raw data on *all* clients share the same structure and are defined on a common feature space. Yet, in practice, data collected by clients may use differing structures: they may not capture the same information, some features may be missing or not stored, or some might have been transformed (*e.g.* via normalization, scaling, or linear combinations). An illustrative example of this, related to Brain-Computer Interfaces (Yger et al., 2016; Lv et al., 2021)

and tackled in this paper, is the scenario where electroencephalography signals are recorded from different subjects, with varying numbers of electrodes and a diverse range of semantic information (*e.g.* motor imagery tasks and resting state). To tackle the challenge of making FL possible in situations where clients have heterogeneous feature spaces – such as disparate dimensionalities or differing semantics of vector coordinates – we present the first personalised FL framework specifically designed to address this learning scenario.

**Proposed Approach.** The key idea of our proposal is driven by two objectives: (i) clients' data have to be embedded in a common latent space, and (ii) data related to the same semantic information (*e.g.* label) have to be embedded in the same region of this latent space. The first objective is a prior necessary step before FL since it allows to define a relevant aggregation scheme on the central server for model parameters (*e.g.* via weighted averaging). The second one is essential for a proper federated learning of the model parameters as FL approaches are known to struggle when data across clients follow different probability distributions (**?**). As shown later, this second objective is not guaranteed by performing client-independent learning of embedding functions, such as via low-dimensional embeddings or autoencoders. To cope with this issue, we align clients' embedded feature distributions with a latent *anchor distribution* that is shared across clients. The learning of the *anchor distribution* happens in a federated way, which means it is updated locally on each client and then combined on a central server through barycenter computation (Veldhuis, 2002; Banerjee et al., 2005). Then, we seamlessly integrate this distribution alignment mechanism, using local embedding functions and anchor distribution, into a personalised federated learning framework that is similar to the approach proposed by Collins et al. (2021), without any loss of generality.

**Contributions.** To help the reader better grasp the differences of our approach with respect to the existing literature, we here spell out our contributions:

1. We are *the first* to formalise the problem of personalised FL on heterogeneous client's feature spaces. In contrast to existing approaches, the proposed general framework, referred to as `FLIC`, allows each client to leverage other clients' data even though they do not have the same raw representation.

2. We introduce a distribution alignment framework and an algorithm that learns the feature embedding functions along with the latent anchor distribution in a local and global federated manner, respectively. We also show how those essential algorithmic components are integrated into a personalised FL algorithm, easing adoption by practitioners.

3. We provide algorithmic and theoretical support to the proposed methodology. In particular, we show that for a simpler but insightful learning scenario, `FLIC` is able to recover the true latent subspace underlying the FL problem.

4. Beyond competitive experimental analyses on toy and real-world problems, we stand out as a pioneer in Brain-Computer Interfaces (BCI) by being the first to learn from heterogeneous BCI datasets using federated learning. The proposed methodology can handle data with different sensor counts and class numbers, a feat not achieved by any other methodology to our knowledge, and can have a strong impact on other medical domains with similar data heterogeneity.

**Conventions and Notations.** The Euclidean norm on $\mathbb{R}^d$ is $\|\cdot\|$. $|S|$ denotes the cardinality of the set S and $\mathbb{N}^* = \mathbb{N} \setminus \{0\}$. For $n \in \mathbb{N}^*$, we refer to $\{1, \ldots, n\}$ with $[n]$. $N(m, \Sigma)$ is the Gaussian distribution with mean vector $m$ and covariance matrix $\Sigma$ and $X \sim \nu$ means that the random variable $X$ is drawn from the probability distribution $\nu$. The Wasserstein distance of order 2 between any probability measures $\mu, \nu$ on $\mathbb{R}^d$ with finite 2-moment is $W_2(\mu, \nu) = (\inf_{\zeta \in \mathcal{T}(\mu, \nu)} \int_{\mathbb{R}^d \times \mathbb{R}^d} \|\theta - \theta'\|^2 d\zeta(\theta, \theta'))^{1/2}$, where $\mathcal{T}(\mu, \nu)$ is the set of transference plans of $\mu$ and $\nu$.

## 2 Related works

As far as our knowledge goes, the proposed methodology is the first one to tackle the problem of FL from heterogeneous feature spaces. However, some related ideas have been proposed in the literature. The idea of using distribution alignement has been considered in the FL literature but only for addressing distribution shifts

Table 1: Related works. PFL refers to horizontal personalised FL, VFL to vertical FL and FTL to federated transfer learning.

| METHOD | TYPE | $\neq$ FEATURE SPACES | MULTI-PARTY | NO SHARED ID | NO SHARED FEATURE |
|---|---|---|---|---|---|
| (Zhang et al., 2021a) | PFL | ✗ | ✓ | ✓ | ✗ |
| (Diao et al., 2021) | PFL | ✗ | ✓ | ✓ | ✗ |
| (Collins et al., 2021) | PFL | ✗ | ✓ | ✓ | ✗ |
| (Shamsian et al., 2021) | PFL | ✗ | ✓ | ✓ | ✗ |
| (Hong et al., 2022) | PFL | ✗ | ✓ | ✓ | ✗ |
| (Makhija et al., 2022) | PFL | ✗ | ✓ | ✓ | ✓ |
| FLIC (THIS PAPER) | PFL | ✓ | ✓ | ✓ | ✓ |
| (Hardy et al., 2017) | VFL | ✓ | ✗ | ✗ | ✓ |
| (Yang et al., 2019) | VFL | ✓ | ✗ | ✗ | ✓ |
| (Gao et al., 2019) | FTL | ✓ | ✓ | ✓ | ✗ |
| (Sharma et al., 2019) | FTL | ✗ | ✗ | ✓ | ✗ |
| (Liu et al., 2020) | FTL | ✓ | ✗ | ✗ | ✓ |
| (Mori et al., 2022) | FTL | ✓ | ✓ | ✗ | ✗ |

on clients (Zhang et al., 2021b; Ye et al., 2022). Other methodological works on autoencoders (Xu et al., 2020), word embeddings (Alvarez-Melis & Jaakkola, 2018; Alvarez-Melis et al., 2019) or FL under high statistical heterogeneity (Makhija et al., 2022; Luo et al., 2021; Zhou et al., 2022) use similar ideas of distribution alignment for calibrating feature extractors and classifiers. Comparing distributions from different spaces has also been considered in a (non-FL) centralised manner using approaches like the Gromov-Wasserstein distance or related distances (Mémoli, 2011; Bunne et al., 2019; Alaya et al., 2022).

Several other works can also be broadly related to the proposed methodology. Loosely speaking, we can divide these related approaches into three categories namely (i) heterogeneous-architecture personalised FL, (ii) vertical FL and (iii) federated transfer learning.

Compared to traditional horizontal personalised FL (PFL) approaches, so-called *heterogeneous-architecture* ones are mostly motivated by local heterogeneity regarding resource capabilities of clients *e.g.* computation and storage (Zhang et al., 2021a; Diao et al., 2021; Collins et al., 2021; Shamsian et al., 2021; Hong et al., 2022; Makhija et al., 2022). Nevertheless, they never consider features defined on heterogeneous subspaces, which is our main motivation. In vertical federated learning (VFL), clients hold disjoint subsets of features. However, a restrictive assumption is that a large number of users are common across the clients (Yang et al., 2019; Hardy et al., 2017; Angelou et al., 2020; Romanini et al., 2021). In addition, up to our knowledge, no vertical personalised FL approach has been proposed so far, which is restrictive if clients have different business objectives and/or tasks. Finally, some works have focused on adapting standard transfer learning approaches with heterogeneous feature domains under the FL paradigm. These *federated transfer learning* (FTL) approaches (Gao et al., 2019; Mori et al., 2022; Liu et al., 2020; Sharma et al., 2019) stand for FL variants of heterogeneous-feature transfer learning where there are $b$ *source* clients and 1 target client with a target domain. However, these methods do not consider the same setting as ours and assume that clients share a common subset of features. We compare the most relevant approaches among the previous ones in Table 1.

## 3 Proposed Methodology

**Problem Formulation.** We consider the problem where $b \in \mathbb{N}^*$ clients want to solve a learning task within the *centralised personalised FL paradigm* (Yang et al., 2019; Kairouz et al., 2021), where a central entity orchestrates the collaborative solving of a common machine learning problem by the $b$ clients, without requiring raw data exchanges. The clients are assumed to possess local data sets $\{D_i\}_{i \in [b]}$ such that, for any $i \in [b]$, $D_i = \{(x_i^{(j)}, y_i^{(j)})\}_{j \in [n_i]}$ where $x_i^{(j)}$ stands for a feature vector, $y_i^{(j)}$ is a label and $n_i = |D_i|$. In contrast

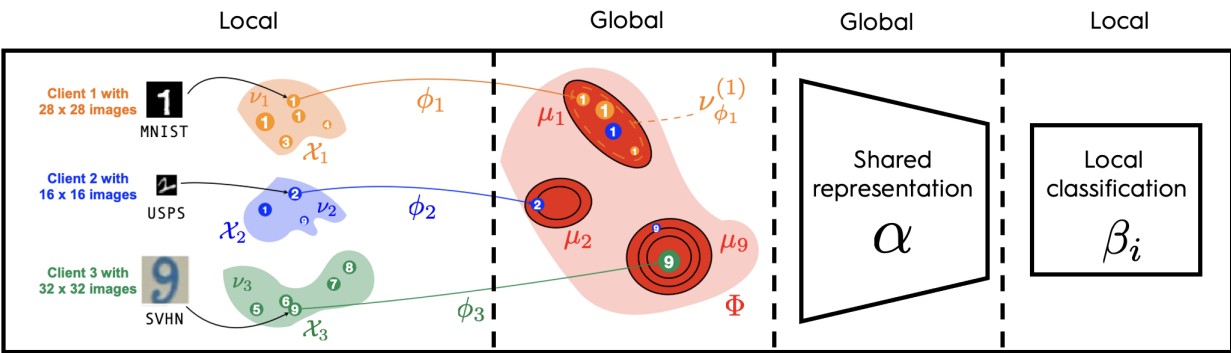

Figure 1: Illustration of part of the proposed methodology for $b = 3$ clients with *heterogeneous* digit images coming from three different data sets namely MNIST (Deng, 2012), USPS (Hull, 1994) and SVHN (Netzer et al., 2011). The circles with digits inside stand for a group of samples, of a given class, owned by a client and the size of the circles indicates their probability mass. In the subspace $\Phi$, $\{\mu_i\}_{i \in [b]}$ (and their level sets) refer to some learnable reference measures to which we seek to align the transformed version $\nu_{\phi_i}$ of $\nu_i$. Personalised FL then occurs in the space $\Phi$ and aims at learning local models $\{\theta_i\}_{i \in [b]}$ for each client as well as $\{\phi_i, \mu_i\}_{i \in [b]}$. Non-personalised FL could also be considered and naturally embedded in the proposed distribution alignement framework.

to existing FL approaches, we assume that the raw input features $\{x_i^{(j)}\}_{j \in [n_i]}$ of clients live in *heterogeneous* spaces *i.e.* for any $i \in [b]$, $x_i^{(j)} \in \mathcal{X}_i$ where $\mathcal{X}_i$ is a client-specific measurable space. More precisely, for any $i \in [b]$ and $j \in [n_i]$, we assume that $x_i^{(j)} \in \mathcal{X}_i \subseteq \mathbb{R}^{k_i}$ such that $\{\mathcal{X}_i\}_{i \in [b]}$ are not part of a common ground metric. This setting is challenging since standard FL approaches (McMahan et al., 2017; Li et al., 2020) and even personalised FL ones (Collins et al., 2021; Hanzely et al., 2021) cannot be directly applied. For simplicity, we assume that all clients want to solve a multi-class classification task with $C \in \mathbb{N}^*$ classes. We discuss later how regression tasks can be encompassed in the proposed framework.

**Methodology.** The goal of the proposed methodology, coined `FLIC`, is to learn a personalised model for each client while leveraging the information stored by other clients' data sets despite the heterogeneity issue. To address this feature space heterogeneity, we propose to map client's features into a fixed-dimension common subspace $\Phi \subseteq \mathbb{R}^k$ by resorting to learned *local* embedding functions $\{\phi_i : \mathcal{X}_i \to \Phi\}_{i \in [b]}$. In order to preserve semantical information (such as the class associated to a feature vector) from the original data distribution, we seek at learning the functions $\{\phi_i\}_{i \in [b]}$ such that they are aligned with (*i.e.* minimise their distance to) some learnable latent anchor distributions that are shared across all clients. These anchor distributions act as universal "calibrators" for clients, preventing similar semantic information from different clients from scattering across the subspace $\Phi$. This scattering would otherwise impede proper subsequent federated learning of the classification model. As depicted in Figure 1, local embedding functions are learned by aligning the mapped probability distributions, denoted as $\nu_{\phi_i}^{(c)}$, conditioned on the class $c \in [C]$, with $C$ learnable anchor measures $\{\mu_c\}_{c \in [C]}$. This alignment is achieved by minimising their distance.

**Remark 1.** *We want to stress the significance of aligning the class-conditional probability distributions $\nu_\phi^{(c)}$ with respect to the anchor distributions $\mu_c$. Local and independent learning of embedding functions $\phi_i$ by each client does not guarantee alignment of resulting probability distributions in the common subspace $\Phi$. As in unsupervised multilingual embeddings (Grave et al., 2019), alignments are crucial for preserving semantic similarity of class information. Misalignment occurs when projecting class-conditionals in a lower-dimensional space using an algorithm, like `t-sne`, that seeks at preserving only local similarities. Indeed, data from different clients are projected in a subspace in which different class-conditionals may overlap. This is also the case when using a neural network with random weights as a projector or an auto-encoder. Examples of such phenomena are illustrated in Figure 3. Notably, this figure shows that the alignment with respect to the anchor distribution is crucial to ensure that the class-conditional distributions are aligned in the common subspace $\Phi$.*

Once data from the heterogeneous spaces are embedded in the same latent subspace $\Phi$, we can deploy a federated learning methodology for training from this novel representation space. While any standard FL approach *e.g.* `FedAvg` (McMahan et al., 2017) can be used, we consider a *personalised* FL where each client has a local model tailored to its specific data distribution as statistical heterogeneities that are still present in $\Phi$ (Tan et al., 2022). Hence, given the aforementioned local embedding functions $\{\phi_i\}$, the model parameters $\{\theta_i \in \mathbb{R}^{d_i}\}$ and some non-negative weights associated to each client $\{\omega_i\}_{i \in [b]}$ such that $\sum_{i=1}^{b} \omega_i = 1$, we consider the following empirical risk minimisation problem:

$$\min_{\theta_{1:b}, \phi_{1:b}} f(\theta_{1:b}, \phi_{1:b}) = \sum_{i=1}^{b} \omega_i f_i(\theta_i, \phi_i) , \tag{1}$$

and for any $i \in [b]$,

$$f_i(\theta_i, \phi_i) = \frac{1}{n_i} \sum_{j=1}^{n_i} \ell \left( y_i^{(j)}, g_{\theta_i}^{(i)} \left[ \phi_i \left( x_i^{(j)} \right) \right] \right) . \tag{2}$$

where $\ell(\cdot, \cdot)$ stands for a classification loss function between the true label $y_i^{(j)}$ and the predicted one $g_{\theta_i}^{(i)}[\phi_i(x_i^{(j)})]$ where $g_{\theta_i}^{(i)}$ is the local model that admits a personalised architecture parameterised by $\theta_i$ and taking as input an embedded feature vector $\phi_i(x_i^{(j)}) \in \Phi$.

**Objective Function.** At this stage, we are able to integrate the FL paradigm and the local embedding function learning into a global objective function we want to optimise, see (1). Remember that we want to learn the parameters $\{\theta_i\}_{i \in [b]}$ of personalised FL models, in conjuction with some local embedding functions $\{\phi_i\}_{i \in [b]}$ and shared anchor distributions $\{\mu_c\}$. In particular, the latter have to be aligned with class-conditional distributions $\{\nu_{\phi_i}^{(c)}\}$. We enforce this alignment via a Wasserstein regularisation term leading us to a regularised version of the empirical risk minimisation problem defined in (1), namely

$$\theta_{1:b}^{\star}, \phi_{1:b}^{\star}, \mu_{1:C}^{\star} = \underset{\theta_{1:b}, \phi_{1:b}, \mu_{1:C}}{\arg\min} \sum_{i=1}^{b} F_i(\theta_i, \phi_i, \mu_{1:C}) ,$$

and for any $i \in [b]$,

$$F_i(\theta_i, \phi_i, \mu_{1:C}) = \omega_i f_i(\theta_i, \phi_i) + \lambda_1 \omega_i \sum_{c \in \mathcal{Y}_i} \mathrm{W}_2^2 \left( \mu_c, \nu_{\phi_i}^{(c)} \right) + \lambda_2 \omega_i \sum_{c \in \mathcal{Y}_i} \frac{1}{J} \sum_{j=1}^{J} \ell \left( c, g_{\theta_i}^{(i)} \left[ Z_c^{(j)} \right] \right) , \tag{3}$$

where $\{Z_c^{(j)}; j \in [J]\}_{c \in [C]}$ stand for samples drawn from $\{\mu_c\}_{c \in [C]}$, and $\lambda_1, \lambda_2 > 0$ are regularisation parameters. The second term in (3) aims at aligning the conditional probability distributions of the transformed features to the anchors. The third one is an optional term aspiring to calibrate the anchor distributions with the classifier in cases where two or more classes are still ambiguous after mapping onto the common feature space; it has also some benefits to tackle covariate shift in standard FL (Luo et al., 2021).

**Design Choices and Justifications.** In the sequel, we consider Gaussian anchor measures $\mu_c = \mathrm{N}(v_c, \Sigma_c)$ where $v_c \in \mathbb{R}^k$ and $c \in [C]$. One of the key advantages of this Gaussian assumption is that, under mild assumptions, it guarantees the existence of a transport map $T^{(i)}$ such that $T_{\#}^{(i)}(\nu_i) = \mu$, owing to Brenier's theorem (Santambrogio, 2015) since a mixture of Gaussians admits a density with respect to the Lebesgue measure. Hence, in our case, learning the local embedding functions boils down to approximating this transport map $T_{\#}^{(i)}$ by $\phi_i$. We also approximate the conditional probability measures $\{\nu_{\phi_i}^{(c)}; c \in \mathcal{Y}_i\}_{i \in [b]}$ by using Gaussian measures $\{\hat{\nu}_{\phi_i}^{(c)} = \mathrm{N}(\hat{m}_i^{(c)}, \hat{\Sigma}_i^{(c)}); c \in \mathcal{Y}_i\}_{i \in [b]}$ such that for any $i \in [b]$ and $c \in [C]$, $\hat{m}_i^{(c)}$ and $\hat{\Sigma}_i^{(c)}$ stand for empirical mean vector and covariance matrix. The relevance of this approximation is detailed in Appendix S1.2.

These two Gaussian choices (for the anchor distributions and the class-conditional distributions) allow us to have a closed-form expression for the Wasserstein distance of order 2 which appears in (3), see *e.g.* Gelbrich (1990); Dowson & Landau (1982). More precisely, we have for any $i \in [b]$ and $c \in [C]$,

$$\mathrm{W}_2^2 \left( \mu_c, \nu_{\phi_i}^{(c)} \right) = \left\| v_c - m_i^{(c)} \right\|^2 + \mathfrak{B}^2 \left( \Sigma_c, \Sigma_i^{(c)} \right) ,$$

Table 2: Current personalised FL techniques that can be embedded in the proposed framework. The parameters $\alpha$ and $\beta_i$ respectively stand for model weights while $\omega \in [0, 1]$. Typically, $\alpha$ stand for the shared weights associated to the shared layers of a neural network architecture and $\beta_i$ for local ones aiming at performing personalised classification.

| Algorithm | Local model | Local weights |
|-----------|-------------|---------------|
| `FedAvg-FT` | $g_{\theta_i}^{(i)} = g_{\theta_i}$ | $\theta_i$ |
| `L2GD` | $g_{\theta_i}^{(i)} = g_{\theta_i}$ | $\theta_i = \omega\alpha + (1-\omega)\beta_i$ |
| `FedRep` | $g_{\theta_i}^{(i)} = g_{\beta_i}^{(i)} \circ g_\alpha$ | $\theta_i = [\alpha, \beta_i]$ |

where $\mathfrak{B}(\cdot, \cdot)$ denotes the Bures distance between two positive definite matrices (Bhatia et al., 2019).

**Remark 2.** *Instead of Gaussian distribution approximations, we can consider more complex probability distributions. For instance, we can use a Gaussian mixture model (GMM) and still be able to compute cheaply the Wasserstein distance (Chen et al., 2018). We can even make no hypothesis on the data distribution and compute the distance for alignment using the linear programming based OT-formulation (Flamary et al., 2021) or use any other IPM such as the maximum mean discrepancy (MMD), see Gretton et al. (2012). However, in practice, we found that the closed-form Wasserstein distance achieves slightly better performance than MMD.*

Regarding the parametrization of the local embedding functions $\phi_i$, we consider a neural network architecture which takes into account the nature of the input data and outputs to a vector of dimension $k$. For instance, for the digit datasets, we consider a convolutional neural network (CNN), followed by pooling and linear layers whereas for the BCI datasets, we have used a fully connected neural network as the data has been preprocessed. The choice of the architecture is left to the user and can be adapted to the problem at hand.

**Reference Distribution for Regression.** For a regression problem, except the change in the loss function, we also need to define properly the reference anchor distributions. Since our goal is to map all samples for all clients into a common latent subspace, in which some structural information about the regression problem is preserved. As such, in order to reproduce the idea of using a Gaussian mixture model as a anchor distribution, we propose to use an infinite number of Gaussian mixtures in which the distribution of a transformed feature vector $\phi(x) \in \mathbb{R}^k$ associated to a response $y$ is going to be mapped on a unit-variance Gaussian distribution whose mean depends uniquely on $y$. Formally, we define the anchor distribution as

$$\mu_y = \mathrm{N}(\mathrm{m}^{(y)}, \mathrm{I}_k)$$

where $\mathrm{m}^{(y)}$ is a vector of dimension $k$ that is uniquely defined. In practice, we consider $\mathrm{m}^{(y)} = ya + (1-y)b$ where $a$ and $b$ are two distinct vectors in $\mathbb{R}^k$.

When training `FLIC`, this means that for a client $i$, we can compute $\mathrm{W}_2^2\left(\mu_y, \nu_{\phi_i}^{(y)}\right)$ based on the set of training samples $\{(x, y)\}$. In practice, if for a given batch of samples we have a single sample of value $x$, then the Wasserstein distance boils to $\|\phi_i(x) - \mathrm{m}^{(y)}\|_2^2$, which means that we are going to map $x$ to its corresponding vector on the segment $[a, b]$.

## 4 Algorithm

As detailed in (3), we perform personalisation under the FL paradigm by considering local model architectures $\{g_{\theta_i}^{(i)}\}_{i \in [b]}$ and local weights $\theta_{1:b}$. As an example, we could resort to federated averaging with fine-tuning (*e.g.* `FedAvg-FT` (Collins et al., 2022)), model interpolation (*e.g.* `L2GD` (Hanzely & Richtárik, 2020; Hanzely et al., 2020)) or partially local models (*e.g.* `FedRep` (Collins et al., 2021) or the works of Oh et al. (2022); Singhal et al. (2021)). Table 2 details how these methods can be embedded into the proposed methodology and explicitly shows the local model and local weights associated to each algorithm as well as how the global weights are taken into account locally. In Algorithm 1, we detail the pseudo-code associated to a specific

instance of the proposed methodology when `FedRep` is resorted to learn model parameters $\{\theta_i\}_{i\in[b]}$ under the FL paradigm. For `FedRep`, each $\theta_i$ includes local and global weights. Besides these learnable parameters, the algorithm also learns the local embedding functions $\phi_{1:b}$ and the anchor distributions $\mu_{1:C}$.

In practice, at a given epoch $t$ of the algorithm, a subset $\mathsf{A}_{t+1} \subseteq [b]$ of clients are selected to participate to the training process. Those clients receive the current latent anchor distribution $\mu_{1:C}^{(t)}$ and the current shared representation part of $\theta^{(t)}$. Then, each client locally updates $\phi_i$, $\theta_i$ and $\mu_{1:C}^{(t)}$. The number of local steps $M$ for each client is a hyperparameter of the algorithm. Afterwards, clients send back to the server their updated version of $\theta^{(t)}$ and $\mu_{1:C}^{(t)}$. Updated global parameters $\theta^{(t+1)}$ and $\mu_{1:C}^{(t+1)}$ are then obtained by weighted averaging of client updates on appropriate manifolds. The use of the Wasserstein loss in (3) naturally leads to perform averaging of the local anchor distributions via a Wasserstein barycenter; algorithmic details are provided in the next paragraph. In Algorithm 1, we use for the sake of simplicity the notation $\texttt{DescStep}(F_i^{(t,m)}, \cdot)$ to denote a (stochastic) gradient descent step on the function $F_i^{(t,m)} = F_i(\theta_i^{(t,m)}, \phi_i^{(t,m)}, \mu_{1:C}^{(t)})$ with respect to a subset of parameters in $(\theta_i, \phi_i, \mu_{1:C})$. This subset is specified in the second argument of `DescStep`. A fully detailed version of Algorithm 1 is provided in the supplementary material (see Algorithm S2).

**Updating and Averaging Anchor Distributions.** In this paragraph, we provide algorithmic details regarding steps 14 and 19 in Algorithm 1. For any $c \in [C]$, the anchor distribution $\mu_c$ involves two learnable parameters namely the mean vector $v_c$ and the covariance matrix $\Sigma_c$. For the mean, step 14 stands for a (stochastic) gradient descent step aiming to obtain a local version of $v_c^{(t)}$ and step 19 boils down to compute the averaging of the mean vectors : $v_c^{(t+1)} = (b/|\mathsf{A}_{t+1}|) \sum_{i\in\mathsf{A}_{t+1}} \omega_i v_{i,c}^{(t+1)}$. For updating the covariance matrix, we resort to the Cholesky decomposition to enforce the positive semi-definite constraint of the covariance matrix. Hence, we rewrite it as $\Sigma_c = L_c L_c^\top$ where $L_c \in \mathbb{R}^{k\times k}$ and optimise in step 14 with respect to the factor $L_c$ instead of $\Sigma_c$. Then, we can handle the gradient computation of the Bures distance in step 14 using the work of Muzellec & Cuturi (2018); and obtain a local factor $L_{i,c}^{(t+1)}$ at iteration $t$. In step 19, we compute $L_c^{(t+1)} = (b/|\mathsf{A}_{t+1}|) \sum_{i\in\mathsf{A}_{t+1}} \omega_i L_{i,c}^{(t+1)}$ and set $\Sigma_c^{(t+1)} = L_c^{(t+1)} [L_c^{(t+1)}]^\top$. When $\lambda_2 = 0$ in (3), these mean vector and covariance matrix updates exactly boil down to perform one stochastic (because of partial participation) gradient descent step to solve the Wasserstein barycenter problem $\arg\min_{\mu_c} \sum_{i=1}^{b} \omega_i W_2^2(\mu_c, \nu_{\phi_i}^{(c)})$.

**Pre-training $\phi_{1:C}$.** Owing to the introduction of reference anchor distributions which carry semantical information, each local feature embedding function can be pre-trained by optimising the loss $\sum_{c\in\mathcal{Y}_i} W_2^2 \left( \mu_c, \nu_{\phi_i}^{(c)} \right)$. While in theory, pre-training may not be necessary, we believe that it helps reach a better solution of the federated learning problem as parameters $\theta_i$ are optimised starting from a better latent representation of $\nu_{\phi_i}$. This is a phenomenon that has been observed in the context of fine-tuning (Kumar et al., 2022) or domain generalisation (Rame et al., 2022). Note that in practice, we initialize the mean vectors $v_c$ of the anchor distributions by randomly sampling a Gaussian distribution with a very large variance. By doing so, we ensure a good separability of the classes and the anchor distributions in the latent space. The covariance matrices are initialized as identity matrices.

**Remark 3.** *Regarding privacy, FL ensures that raw data never leaves the client device. Compared to the base FL algorithm such as `FedRep`, `FLIC` exchanges between clients and server the anchor distribution $\mu_{i,1:C}^{(t+1)}$. This can be seen a further privacy risk. However, the anchor distribution is a set of $C$ Gaussian distributions, providing only aggregated information about the data and it needs to be clarified based on further work the level of privacy it brings.. A possible solution to mitigate potential privacy leak is to consider differential privacy (Dwork & Roth, 2014) and use a differentially private version of the Wasserstein barycenter algorithm based on a differentially private Wasserstein distance (Lê Tien et al., 2019; Goldfeld & Greenewald, 2020; Rakotomamonjy & Ralaivola, 2021).*

## 5 Non-Asymptotic Convergence Guarantees in a Simplified Setting

Deriving non-asymptotic convergence bounds for Algorithm 1 in the general case is challenging since the considered $C$-class classification problem leads to jointly solving personalised FL and federated Wasserstein barycenter problems. Regarding the latter, obtaining non-asymptotic convergence results is still an active research area in the centralised learning framework (Altschuler et al., 2021). As such, we propose to analyse

---

**Algorithm 1** `FLIC` for `FedRep`

---

**Require:** initialisation $\mu_{1:C}^{(0)}$, $\phi_{1:b}^{(0,0)}$, $\theta^{(0)} = [\alpha^{(0)}, \beta_{1:b}^{(0,0)}]$.

1: **for** $t = 0$ **to** $T - 1$ **do**
2:     Sample a set of $\mathsf{A}_{t+1}$ of active clients.
3:     **for** $i \in \mathsf{A}_{t+1}$ **do**
4:         *// Communication to clients*
5:         The central server sends the global parameters $\alpha^{(t)}$ and $\mu_{1:C}^{(t)}$ to $\mathsf{A}_{t+1}$ to clients.
6:         *// Update local parameters*
7:         **for** $m = 0$ **to** $M - 1$ **do**
8:             $\phi_i^{(t,m+1)} \leftarrow \texttt{DescStep}\left(F_i^{(t,m)}, \phi_i^{(t,m)}\right)$.
9:             $\beta_i^{(t,m+1)} \leftarrow \texttt{DescStep}\left(F_i^{(t,m)}, \beta_i^{(t,m)}\right)$.
10:       $\phi_i^{(t+1,0)} = \phi_i^{(t,M)}$.
11:       $\beta_i^{(t+1,0)} = \beta_i^{(t,M)}$.
12:       *// Update global parameters*
13:       $\alpha_i^{(t+1)} \leftarrow \texttt{DescStep}\left(F_i^{(t,M)}, \alpha^{(t)}\right)$.
14:       $\mu_{i,1:C}^{(t+1)} \leftarrow \texttt{DescStep}\left(F_i^{(t,M)}, \mu_{1:C}^{(t)}\right)$.
15:       *// Communication with the server*
16:       Send $\alpha_i^{(t+1)}$ and $\mu_{i,1:C}^{(t+1)}$ to central server.
17:     *// Averaging global parameters*
18:     $\alpha^{(t+1)} = \frac{b}{|\mathsf{A}_{t+1}|} \sum_{i \in \mathsf{A}_{t+1}} w_i \alpha_i^{(t+1)}$
19:     $\mu_{1:C}^{(t+1)} \leftarrow \texttt{WassersteinBarycenter}(\{\mu_{i,1:C}^{(t+1)}\})$

**Ensure:** parameters $\alpha^{(T)}$, $\mu_{1:C}^{(T)}$, $\phi_{1:b}^{(T,0)}$, $\beta_{1:b}^{(T,0)}$.

---

a simpler regression framework where the anchor distribution is known beforehand and not learned under the FL paradigm. While we acknowledge that this theoretical analysis is based on a simplified setting of our approach, it still offers insightful perspective and we leave the general case for future work.

More precisely, we assume that $x_i^{(j)} \sim \mathrm{N}(m_i, \Sigma_i)$ with $m_i \in \mathbb{R}^{k_i}$ and $\Sigma_i \in \mathbb{R}^{k_i \times k_i}$ for $i \in [b], j \in [n_i]$. In addition, we consider that the continuous scalar labels are generated via the oracle model $y_i^{(j)} = (A^\star \beta_i^\star)^\top \phi_i^\star(x_i^{(j)})$ where $A^\star \in \mathbb{R}^{k \times d}$, $\beta_i^\star \in \mathbb{R}^d$ and $\phi_i^\star(\cdot)$ are the ground-truth parameters and the feature embedding functions, respectively. We make the following assumptions on the ground truth, which are inherited from those of `FedRep`.

**H1.** *(i) For any $i \in [b]$, $k_i \geq k$.*

*(ii) For any $i \in [b]$, $j \in [n_i]$, embedded features $\phi_i^\star(x_i^{(j)})$ are distributed according to $\mathrm{N}(0_k, \mathrm{I}_k)$.*

*(iii) Ground-truth model parameters satisfy $\|\beta_i^\star\|_2 = \sqrt{d}$ for $i \in [b]$ and $A^\star$ has orthonormal columns.*

*(iv) For any $t \in \{0, \ldots, T-1\}$, $|\mathsf{A}_{t+1}| = b'$ with $1 \leq b' \leq b$, and if we select $b'$ clients, their ground-truth head parameters $\{\beta_i^\star\}_{i \in \mathsf{A}_{t+1}}$ span $\mathbb{R}^d$.*

*(v) In (2), $\ell(\cdot, \cdot)$ is the $\ell_2$ norm, $\omega_i = 1/b$, $\theta_i = [A, \beta_i]$ and $g_{\theta_i}^{(i)}(x) = (A\beta_i)^\top x$ for $x \in \mathbb{R}^k$.*

Under **H**1-(i), Delon et al. (2022, Theorem 4.1) show that $\phi_i^\star$ can be expressed as a non-unique affine map with closed-form expression. To align with the true latent anchor distribution $\mu = \mathrm{N}(0_k, \mathrm{I}_k)$, we propose to estimate $\hat{\phi}_i$ by leveraging this closed-form mapping between $\mathrm{N}(m_i, \Sigma_i)$ and $\mu$. Because of the non-unicity of $\phi_i^\star$, we show in Theorem 1 that we can only recover it up to a matrix multiplication. Interestingly, Theorem 1 also proves that the global representation $A^{(T)}$ learnt via `FedRep` (see Algorithm S3 in Appendix) is able to correct this feature mapping indetermination. Associated convergence behavior is illustrated in Figure 2 on a toy example whose details are postponed to Appendix S2.

**Theorem 1.** *Assume **H**1. Then, for any $x_i \in \mathbb{R}^{k_i}$, we have $\hat{\phi}_i(x_i) = Q\phi_i^\star(x_i)$ where $Q \in \mathbb{R}^{k \times k}$ is of the form* $\mathrm{diag}_k(\pm 1)$. *Under additional technical assumptions detailed in Appendix S2, we have for any $t \in \{0, \ldots, T-1\}$*

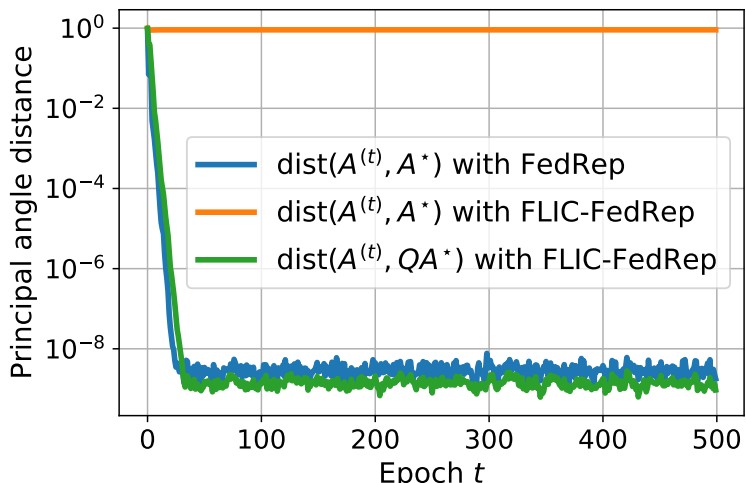

Figure 2: Illustration of the convergence behavior of `FLIC` on a toy linear regression problem. The plot shows the principal angle distance between the estimated linear global map $A^{(t)}$ and the true linear global map $A^\star$ (up to a matrix multiplication). (blue) $A^{(t)}$ is estimated based on the latent space feature mapped obtained using $\phi_i^\star$ and `FedRep`. (orange) $A^{(t)}$ is estimated based on the latent space feature mapped obtained using $\hat{\phi}_i$. (green) $A^{(t)}$ is estimated based on the latent space feature mapped obtained using $\hat{\phi}_i$ but compared to $QA^\star$. Here, we show `FLIC` is able to recover the true global map $A^\star$ up to a matrix multiplication.

*and with high probability,*

$$\text{dist}(A^{(t+1)}, QA^\star) \leq (1 - \kappa)^{(t+1)/2} \text{dist}(A^{(0)}, QA^\star),$$

*where $\kappa \in (0, 1)$ is detailed explicitly in Theorem S3 and* dist *denotes the principal angle distance.*

## 6  Numerical Experiments

For numerically validating the benefits associated to the proposed methodology `FLIC`, we consider toy problems with different characteristics of heterogeneity; as well as experiments on real data, namely (i) a digit classification problem from images of different sizes, (ii) an object classification problem from either images or text captioning on clients, and (iii) a Brain-Computer Interfaces problem. Code for reproducing part of the experiments is available at https://github.com/arakotom/flic.

**Baselines.** Since the problem we are addressing is novel, no FL competitor in the literature can serve as a baseline beyond local learning. However, we propose to modify the methodology proposed by Makhija et al. (2022) and Collins et al. (2022) to make them applicable to clients with heterogeneous feature spaces. The method proposed by Makhija et al. (2022) handles local representation models with different architectures. Since their key idea is to align the latent representations of fixed-dimensionality inputs shared by the server to all clients, we propose an alternative approach called `FedHeNN`, that works for clients with different feature spaces, where we build a Representation Alignment Dataset (RAD) based on the largest feature space and then prune it to obtain a lower-dimensional RAD for each client. We can also adapt the `FedRep` approach (Collins et al., 2022) to our setting by considering a local feature transformation followed by a shared global representation model and a local classification head. This approach, denoted as `HetFedRep`, maps the input data to fixed dimensionality before the shared global representation model. We can understand this approach as a special case of `FLiC` where the local feature transformation are not enforced to align with the anchor distributions. We adapted our network architecture to match the baselines by considering two variants. Following Makhija et al. (2022), we treated all layers except the last as the representation learning module for a fair comparison. Therefore, in our approach, the alignment applies to the penultimate layer, and the last layer is the classifier layer. We call this model `FLIC`-Class. Additionally, we introduced another model, called

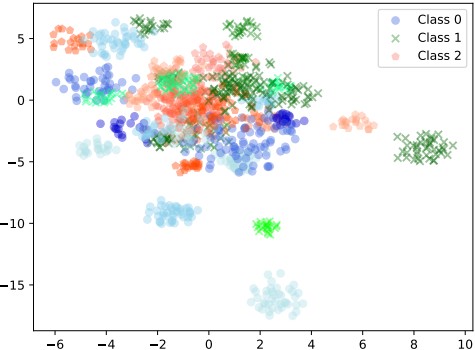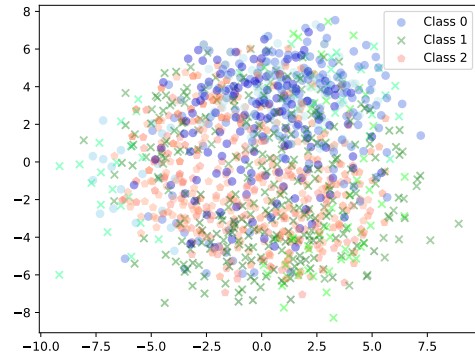

Figure 3: Projecting the Gaussian class-conditionals stored on 20 clients obtained from the "added features" toy problem. For readability, the problem has been made easier by considering only 3 classes with all classes on clients. Means of the class-conditionals are drawn from a zero-mean Gaussian with a covariance matrix $\sigma^2\mathbf{I}$ with $\sigma = 3$. Added features range from 1 to 10. (left) *t-sne* projection. (right) *Multi Dimensional Scaling* (MDS) projection. Different tones of a color represent the same class-conditionals of a given problem. From this figure, we remark the overlapping of classes regardless of the algorithm used for projection (*t-sne* uses a random initialization and while *MDS* uses a deterministic ones). This emphasizes the need for a proper alignement of the class-conditionals during projection.

FLIC-HL, similar to FedRep, but with an extra trainable global hidden layer with $\alpha$ and $\beta_i$ as parameters for respectively the shared representation and classification layers.

**Data Sets.** We consider four different classification problems to assess the performances of our approach. For all simulations, we assume prior probability shift *e.g* each client will have access to data of only specific classes. The first problem is a toy problem with 20 classes and Gaussian class-conditional distributions, where we conduct two sub-experiments: adding random spurious features and applying a random linear mapping, both of random dimensionality on each client. Section S3.1 in the supplementary material provides more details about these data sets. The second problem involves digit classification using MNIST and USPS datasets, with dimensions of $28 \times 28$ and $16 \times 16$, respectively. Each client hosts a subset of either MNIST or USPS dataset. The third experiment addresses a multimodal problem using a subset of the *TextCaps* dataset (Sidorov et al., 2020), an image captioning dataset. We converted it into a 4-class classification problem with $12,000$ and $3,000$ examples for training and testing, respectively, based on caption text or image. We used pre-trained models (Bert and ResNet) to embed the caption and image into 768-dimensional and 512-dimensional vectors. To create heterogeneity, we randomly pruned 10% of features on each client. Each client hosts either image or text embeddings. Finally, the fourth problem is a real medical problem denoted as Brain-Computer Interface (BCI) which consists in classifying mental imagery EEG datasets into five classes. The datasets we have considered is based on six datasets from the mental imagery MOABB data repository (Jayaram & Barachant, 2018) (details are given in Section S3.1 in the supplement). Each of those EEG datasets have been acquired on different subjects, have different number of channels and classes. We used a vectorised channel-dependent covariance matrices representations of each EEG signals as a feature (Yger et al., 2016; Barachant et al., 2022). Hence, the dimensionality of the feature space is different for each dataset. We have considered each subject in all the experiments as a client owning his own dataset. In practice, the number of training examples on client ranges from 30 to 600 while the dimensionality of the features goes from 6 to 1,830.

**Illustrating the need for anchor distributions.** The main bottleneck for applying FL algorithms to heterogeneous feature spaces is the lack of a common space. However, one can argue that this common space can be created by projecting the data onto a joint common space. As we have claimed, we illustrated here that this is not sufficient. To do so, we have considered the "added noisy features" toy problem with

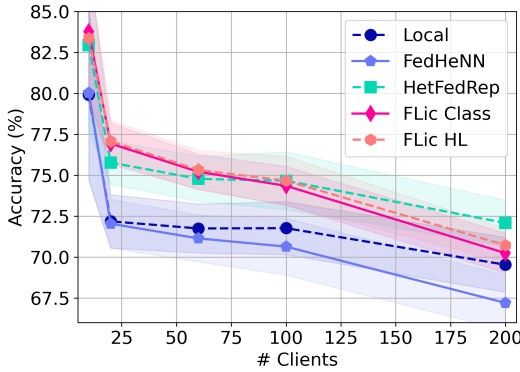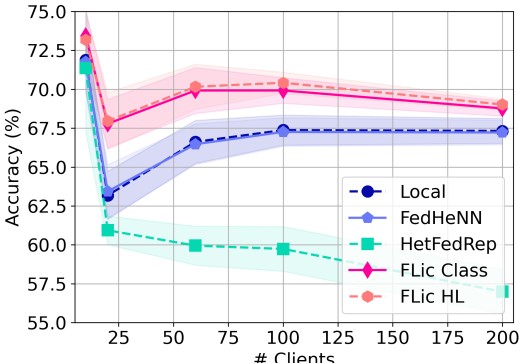

Figure 4: Performance of `FLIC` and competitors on the toy data sets with respect to the number of clients. (left) Gaussian classes in dimension $k = 5$ with added noisy feature. (right) Gaussian classes in dimension $k = 30$, transformed by a random linear mapping. Only 3 classes are present on each client among the 20 possible ones.

the 3 class-conditionals over 20 clients. We have projected those class-conditionals onto a common space using different projection algorithms, namely *t-sne* (Van der Maaten & Hinton, 2008) and multi-dimensional scaling (*MDS*) (Kruskal, 1964). The results are shown in Figure 3. We can note that just projecting into a common subspace without taking into account the semantic of the class-conditionals leads to overlapping of classes. This emphasizes the need for a proper alignment of the class-conditionals during projection, based on reference distribution as we propose in `FLIC`. In Appendix S3.7, we provide quantitative evidence of the need for a proper alignment of the class-conditionals by exhibiting the poor performance of `FedRep` trained on data from this common space.

**Experimental Setting.** For the experimental analysis, we use the codebase of Collins et al. (2021) with some modifications to meet our setting. For all experiments, we consider $T = 50$ communication rounds for all algorithms; and at each round, a client participation rate of $r = 0.1$. The number of local epochs for training has been set to $M = 10$. As optimisers, we have used an Adam optimiser with a learning rate of 0.001 for all problems and approaches. Further details are given in Section S3.3 in the supplement. For each component of the latent anchor distribution, we consider a Gaussian with learnable mean vectors and fixed Identity covariance matrix. As such, the Wasserstein barycenter computation boils down to simply average the mean of client updates and for computing the third term in (3), we just sample from the Gaussian distribution. Accuracies are computed as the average accuracy over all clients after the last epoch in which all local models are trained.

**Results on Toy Data Sets.** Figure 4 depicts the performance, averaged over 5 runs, of the different algorithms with respect to the number of clients and when only 3 classes are present in each client. Since the fixed amount of training data (2000 per classes) are equally shared across clients holding a given class, as we increase the number of clients, we expect the problem to be harder and the performance to decrease. For both data sets, we can note that for the *noisy feature* setting, `FLIC` improves on FedHeNN of about 3% of accuracy across the setting, performs better than local learning and is comparable to HetFedRep. For the *linear mapping* setting, `FLIC` achieves better than other approaches with a gain of performance of about 4% while the gap tends to decrease as the number of clients increases. For this problem, HetFedRep fails to achieve good performance, highlighting the need for a proper alignment of the class-conditionals. Since the local embedding functions are handled as any local layers and not enforced to align with the anchor distributions. Interestingly for both problems, `FLIC`-HL performs slightly better than `FLIC`-Class showing the benefit of adding a shared representation layer $\alpha$ in addition to the local embedding functions and the classification layer.

Figure 5 also illustrates how samples embedded with $\phi_i$ evolve during training towards the anchor distributions $\mu_{1:C}$. At start, they are clustered client-wise (same marker with different colors are grouped together.) As

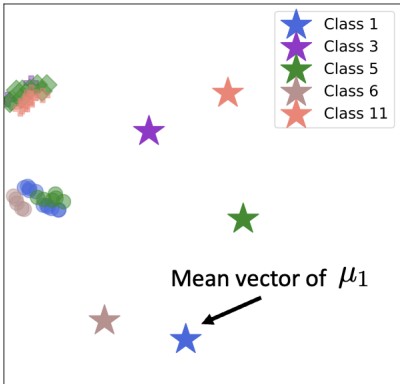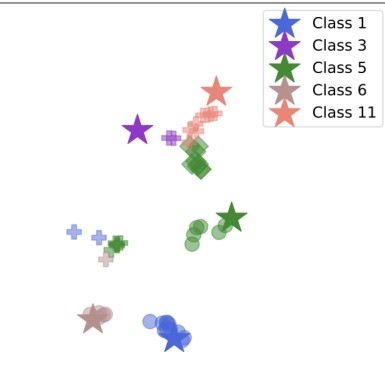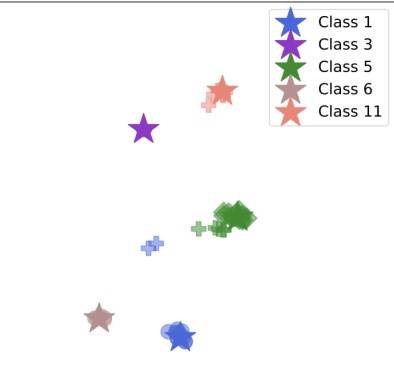

Figure 5: . 2D *t-sne* projection of 5 classes partially shared by 3 clients for the **toy linear mapping** dataset after learning the local transformation functions for (left) 10 epochs, (middle) 50 epochs, (right) 100 epochs. The three different markers represent the different clients while classes are represented by different color tones. The ⋆ marker represents the class-conditional mean of the reference anchor distribution. We note that training set converges towards those means.

Table 3: Performance over 3 runs of our `FLIC` model and the competitors on some real-data problems (*Digits* and *TextCaps* data set).

| Data sets (setting) | Local | FedHeNN | HetFedRep | FLIC-Class | FLIC-HL |
|---|---|---|---|---|---|
| Digits ($b = 100$, 3 Classes/client) | $97.49 \pm 0.4$ | $97.45 \pm 0.5$ | $57.85 \pm 1.4$ | $\mathbf{97.83 \pm 0.3}$ | $97.70 \pm 0.2$ |
| Digits ($b = 100$, 5 Classes/client) | $96.15 \pm 0.3$ | $96.15 \pm 0.2$ | $54.87 \pm 5.0$ | $96.46 \pm 0.6$ | $\mathbf{96.54 \pm 0.7}$ |
| Digits ($b = 200$, 3 Classes/client) | $93.33 \pm 0.2$ | $93.40 \pm 0.4$ | $67.99 \pm 2.2$ | $94.50 \pm 0.3$ | $\mathbf{94.51 \pm 0.3}$ |
| Digits ($b = 200$, 5 Classes/client) | $87.48 \pm 1.5$ | $87.22 \pm 1.8$ | $48.88 \pm 3.0$ | $\mathbf{91.11 \pm 0.6}$ | $91.10 \pm 0.7$ |
| TextCaps ($b = 100$, 2 Classes/client) | $84.19 \pm 0.8$ | $83.99 \pm 0.7$ | $87.05 \pm 0.7$ | $89.14 \pm 1.1$ | $\mathbf{89.68 \pm 0.7}$ |
| TextCaps ($b = 100$, 3 Classes/client) | $76.04 \pm 0.8$ | $75.39 \pm 0.9$ | $77.99 \pm 0.6$ | $81.27 \pm 0.2$ | $\mathbf{81.50 \pm 0.2}$ |
| TextCaps ($b = 200$, 2 Classes/client) | $83.78 \pm 1.8$ | $83.89 \pm 1.7$ | $85.48 \pm 1.5$ | $87.73 \pm 0.8$ | $\mathbf{87.74 \pm 1.3}$ |
| TextCaps ($b = 200$, 3 Classes/client) | $74.95 \pm 1.1$ | $74.77 \pm 1.0$ | $75.73 \pm 0.8$ | $\mathbf{79.08 \pm 0.7}$ | $78.49 \pm 0.7$ |
| BCI (b=54) | $73.51 \pm 0.8$ | $70.84 \pm 1.0$ | $75.03 \pm 0.6$ | $75.17 \pm 0.9$ | $\mathbf{76.27 \pm 0.2}$ |
| BCI (b=40) | $73.98 \pm 0.2$ | $71.48 \pm 0.6$ | $74.23 \pm 0.7$ | $75.09 \pm 1.0$ | $\mathbf{75.82 \pm 0.3}$ |

iterations go, samples converge towards the relevant class wise anchor distributions. For instance, we note that after 50 epochs, samples from the class 5 are converging towards the mean of the related anchor distribution. After 100 epochs, they are almost all concentrated around the mean.

**Results on Digits and TextCaps Data Sets.** Performance, averaged over 3 runs, of all algorithms on the real-word problems are reported in Table 3. For the *Digits* data set problem, we first remark that in all situations, FL algorithms performs a bit better than local learning. In addition, both variants of `FLIC` achieve better accuracy than competitors. Difference in performance in favor of `FLIC` reaches 3% for the most difficult problem. For the *TextCaps* data set, gains in performance of `FLIC`-HL reach about 4% across settings. While `FedHeNN` and `FLIC` algorithms follow the same underlying principle (alignment of representation in a latent space), our framework benefits from the use of the latent anchor distributions, avoiding the need of sampling from the original space. Instead, `FedHeNN` may fail as the sampling strategy of their RAD approach suffers from the curse of dimensionality and does not properly lead to a successful feature alignment.

**Results on Brain-Computer interfaces.** For the BCI problem, performances are in Table 3. In this case, the `Local` model performance corresponds also to the usual BCI performance measure as models are usually subject-specific. We can note that `FLIC`-HL achieves better performance than all competitors with a gain of

Table 4: Performance over 3 runs of our `FLIC` model and the competitors on the same real-world problems when processed so as to have same input sizes.

| Data sets (setting) | Local | FedHeNN | HetFedRep | FedRep | FLIC-Class | FLIC-HL |
|---|---|---|---|---|---|---|
| Digits-Resize ($b = 100$, 3 Classes) | $97.70 \pm 0.1$ | $97.62 \pm 0.1$ | $92.63 \pm 1.8$ | $95.76 \pm 0.4$ | $\mathbf{98.14 \pm 0.1}$ | $98.05 \pm 0.1$ |
| Digits-Resize ($b = 100$, 5 Classes) | $96.36 \pm 0.1$ | $96.37 \pm 0.2$ | $94.90 \pm 0.7$ | $96.07 \pm 0.5$ | $\mathbf{96.94 \pm 0.1}$ | $96.91 \pm 0.1$ |
| Digits-Resize ($b = 200$, 3 Classes) | $93.62 \pm 0.3$ | $93.56 \pm 0.2$ | $69.21 \pm 3.0$ | $92.93 \pm 0.8$ | $\mathbf{94.73 \pm 0.4}$ | $94.54 \pm 0.2$ |
| Digits-Resize ($b = 200$, 5 Classes) | $87.74 \pm 1.3$ | $87.49 \pm 1.0$ | $69.27 \pm 1.8$ | $\mathbf{94.57 \pm 0.9}$ | $91.40 \pm 0.4$ | $91.16 \pm 0.3$ |
| TextCaps-Clip ($b = 100$, 2 Classes) | $96.55 \pm 0.5$ | $96.44 \pm 0.5$ | $96.45 \pm 0.4$ | $82.31 \pm 2.6$ | $96.59 \pm 0.5$ | $\mathbf{96.65 \pm 0.4}$ |
| TextCaps-Clip ($b = 100$, 3 Classes) | $\mathbf{94.38 \pm 0.2}$ | $94.21 \pm 0.3$ | $94.13 \pm 0.2$ | $76.47 \pm 2.7$ | $94.34 \pm 0.3$ | $94.21 \pm 0.3$ |
| TextCaps-Clip ($b = 200$, 2 Classes) | $96.27 \pm 0.4$ | $96.16 \pm 0.4$ | $96.36 \pm 0.2$ | $83.33 \pm 1.7$ | $\mathbf{96.55 \pm 0.2}$ | $96.44 \pm 0.2$ |
| TextCaps-Clip ($b = 200$, 3 Classes) | $93.78 \pm 0.2$ | $93.55 \pm 0.2$ | $93.74 \pm 0.4$ | $73.23 \pm 0.5$ | $\mathbf{94.04 \pm 0.2}$ | $93.88 \pm 0.3$ |
| BCI-common (b=40) | $71.24 \pm 0.6$ | $70.63 \pm 0.4$ | $72.09 \pm 0.1$ | $71.50 \pm 0.3$ | $\mathbf{72.17 \pm 0.3}$ | $72.12 \pm 0.6$ |

about 3% of accuracy compared to BCI baseline. In addition, we pave the way to learning BCI models with heterogeneous datasets.

**Preprocessing datasets to same dimensionality.** Some preprocessing can be applied to the above datasets so that standard "same dimensionality" FL methods can be considered. We can apply a simple resizing of an image. For the TextCaps classification problem we can extract features from multimodal embedder such as CLIP Radford et al. (2021b) in which embeddings between text and images are aligned and of dimension 512. For the BCI problem, we used EEGs acquired from the set of common electrodes. This experiment also serves at proving that `FLIC` is not only useful for heterogeneous feature spaces but can be applied when feature spaces are identical. In such a case, we expect to benefit from the alignment of the class-conditionals to enhance performances. We have run the same experiments as above but using these preprocessings and also compared to plain `FedRep` and report the results in Table 4. `FLIC` still achieves slightly better performance than competitors on 7 out of 9 settings, but more importantly, one should highlight the gain in performance on BCI problems when considering all sensors at disposal.

## 7 Conclusion

We have introduced a novel and general framework, referred to as `FLIC`, for personalised FL when clients have heterogeneous feature spaces. Under this framework, we proposed a FL algorithm involving two key components: (i) a local feature embedding function; and (ii) a latent anchor distribution which allows to match similar semantic information from each client. Experiments on relevant data sets have shown that `FLIC` achieves better performances than competing approaches. Finally, we provided theoretical support to the proposed methodology, notably via a non-asymptotic convergence result.

**Limitations and Broader impacts of `FLIC`.** One main limitation of `FLIC` is that it requires a common feature space to be defined with an *ad-hoc* dimensionality. While this dimensionality can be chosen by the user, it is not clear how to select it in practice and has to be set to default value (in our case 64). In addition, the proposed approach has a computational overhead due to the need of learning the local embedding functions. In addition, it is worth noting that the proposed approach is a stateful algorithm that is applicable in a cross-silo setting. `FLIC` has the potential to widen the scope of privacy-aware FL applications by allowing clients to have heterogeneous feature spaces. This is particularly relevant for medical applications where data are collected from different sources and may have different formats.

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

# Appendices

## Personalised Federated Learning On Heterogeneous Feature Spaces

**Notations and conventions.** We denote by $\mathcal{B}(\mathbb{R}^d)$ the Borel $\sigma$-field of $\mathbb{R}^d$, $\mathbb{M}(\mathbb{R}^d)$ the set of all Borel measurable functions $f$ on $\mathbb{R}^d$ and $\|\cdot\|$ the Euclidean norm on $\mathbb{R}^d$. For $\mu$ a probability measure on $(\mathbb{R}^d, \mathcal{B}(\mathbb{R}^d))$ and $f \in \mathbb{M}(\mathbb{R}^d)$ a $\mu$-integrable function, denote by $\mu(f)$ the integral of $f$ with respect to (w.r.t.) $\mu$. Let $\mu$ and $\nu$ be two sigma-finite measures on $(\mathbb{R}^d, \mathcal{B}(\mathbb{R}^d))$. Denote by $\mu \ll \nu$ if $\mu$ is absolutely continuous w.r.t. $\nu$ and $\mathrm{d}\mu/\mathrm{d}\nu$ the associated density. We say that $\zeta$ is a transference plan of $\mu$ and $\nu$ if it is a probability measure on $(\mathbb{R}^d \times \mathbb{R}^d, \mathcal{B}(\mathbb{R}^d \times \mathbb{R}^d))$ such that for all measurable set $\mathsf{A}$ of $\mathbb{R}^d$, $\zeta(\mathsf{A} \times \mathbb{R}^d) = \mu(\mathsf{A})$ and $\zeta(\mathbb{R}^d \times \mathsf{A}) = \nu(\mathsf{A})$. We denote by $\mathcal{T}(\mu, \nu)$ the set of transference plans of $\mu$ and $\nu$. In addition, we say that a couple of $\mathbb{R}^d$-random variables $(X, Y)$ is a coupling of $\mu$ and $\nu$ if there exists $\zeta \in \mathcal{T}(\mu, \nu)$ such that $(X, Y)$ are distributed according to $\zeta$. We denote by $\mathcal{P}_1(\mathbb{R}^d)$ the set of probability measures with finite 1-moment: for all $\mu \in \mathcal{P}_1(\mathbb{R}^d)$, $\int_{\mathbb{R}^d} \|x\| \mathrm{d}\mu(x) < \infty$. We denote by $\mathcal{P}_2(\mathbb{R}^d)$ the set of probability measures with finite 2-moment: for all $\mu \in \mathcal{P}_2(\mathbb{R}^d)$, $\int_{\mathbb{R}^d} \|x\|^2 \mathrm{d}\mu(x) < \infty$. We define the squared Wasserstein distance of order 2 associated with $\|\cdot\|$ for any probability measures $\mu, \nu \in \mathcal{P}_2(\mathbb{R}^d)$ by

$$
\mathrm{W}_2^2(\mu, \nu) = \inf_{\zeta \in \mathcal{T}(\mu, \nu)} \int_{\mathbb{R}^d \times \mathbb{R}^d} \|x - y\|^2 \mathrm{d}\zeta(x, y) \,.
$$

By Villani (2008, Theorem 4.1), for all $\mu$, $\nu$ probability measures on $\mathbb{R}^d$, there exists a transference plan $\zeta^\star \in \mathcal{T}(\mu, \nu)$ such that for any coupling $(X, Y)$ distributed according to $\zeta^\star$, $\mathrm{W}_2(\mu, \nu) = \mathbb{E}[\|x - y\|^2]^{1/2}$. This kind of transference plan (respectively coupling) will be called an optimal transference plan (respectively optimal coupling) associated with $W_2$. By Villani (2008, Theorem 6.16), $\mathcal{P}_2(\mathbb{R}^d)$ equipped with the Wasserstein distance $\mathrm{W}_2$ is a complete separable metric space. For the sake of simplicity, with little abuse, we shall use the same notations for a probability distribution and its associated probability density function. For $n \geq 1$, we refer to the set of integers between 1 and $n$ with the notation $[n]$. The $d$-multidimensional Gaussian probability distribution with mean $\mu \in \mathbb{R}^d$ and covariance matrix $\Sigma \in \mathbb{R}^{d \times d}$ is denoted by $\mathrm{N}(\mu, \Sigma)$. Given two matrices $M, N \in \mathbb{R}^{k \times d}$, the principal angle distance between the subspaces spanned by the columns of $M$ and $N$ is given by $\mathrm{dist}(M, N) = \|\hat{M}_\perp^\top \hat{N}\|_2 = \|\hat{N}_\perp^\top \hat{M}\|_2$ where $\hat{M}, \hat{N}$ are orthonormal bases of $\mathrm{Span}(M)$ and $\mathrm{Span}(N)$, respectively. Similarly, $\hat{M}_\perp, \hat{N}_\perp$ are orthonormal bases of orthogonal complements $\mathrm{Span}(M)^\perp$ and $\mathrm{Span}(N)^\perp$, respectively. This principal angle distance is upper bounded by 1, see Jain et al. (2013, Definition 1). **Outline.** This supplementary material aims at providing the interested reader with a further understanding of the statements pointed out in the main paper. More precisely, in Appendix S1, we support the proposed methodology `FLIC` with algorithmic and theoretical details. In Appendix S2, we prove the main results stated in the main paper. Finally, in Appendix S3, we provide further experimental design choices and show complementary numerical results.

## S1 Algorithmic and Theoretical Insights

In this section, we highlight alternative but limited ways to cope with feature space heterogeneity; and justify the usage, in the objective function (3) of the main paper, of Wasserstein distances with empirical probability distributions instead of true ones. In addition, we detail the general steps depicted Algorithm 1.

### S1.1 Some Limited but Common Alternatives to Cope with Feature Space Heterogeneity

Depending on the nature of the spaces $\{\mathcal{X}_i\}_{i \in [b]}$, the feature transformation functions $\{\phi_i\}_{i \in [b]}$ can be either known beforehand or more difficult to find. As an example, if for any $i \in [b]$, $\mathcal{X}_i \subseteq \mathcal{X}$, then we can set mask functions as feature transformation functions in order to only consider features that are shared across all the clients. Besides, we could consider multimodal embedding models to perform feature transformation

on each client (Duquenne et al., 2021). For instance, if clients own either pre-processed images or text of titles, descriptions and tags, then we can use the Contrastive Language-Image Pre-Training (CLIP) model as feature transformation function (Radford et al., 2021a). These two examples lead to the solving of a classical (personalised) FL problem which can be performed using existing state-of-the-art approaches. However, when the feature transformation functions cannot be easily found beforehand, solving the FL problem at stake becomes more challenging and has never been addressed in the federated learning literature so far, up to the authors' knowledge.

### S1.2 Use of Wasserstein Losses Involving Empirical Probability Distributions

Since the true probability distributions $\{\nu^{(c)}_{\phi_i}; c \in \mathcal{Y}_i\}_{i \in [b]}$ are unknown a priori, we propose in the main paper to estimate the latter using $\{\hat{\nu}^{(c)}_{\phi_i}; c \in \mathcal{Y}_i\}_{i \in [b]}$ and to replace $\mathrm{W}^2_2\left(\mu_c, \nu^{(c)}_{\phi_i}\right)$ by $\mathrm{W}^2_2\left(\mu_c, \hat{\nu}^{(c)}_{\phi_i}\right)$ in the objective function (3) in the main paper. As shown in the following result, this assumption is theoretically grounded when the marginal distributions of the input features are Gaussian.

**Theorem S2.** *For any $i \in [b]$ and $c \in [C]$, let $n^{(c)}_i = |\mathrm{D}^{(c)}_i|$ where $\mathrm{D}^{(c)}_i$ denotes the subset of the local data set $\mathrm{D}_i$ only involving observations associated to the label $c$. Besides, assume that $\nu^{(c)}_{\phi_i}$ is Gaussian with mean vector $m^{(c)}_i \in \mathbb{R}^k$ and full-rank covariance matrix $\Sigma^{(c)}_i \in \mathbb{R}^{k \times k}$. Then, we have in the limiting case $n^{(c)}_i \to \infty$,*

$$\sqrt{n^{(c)}_i}\left(\mathrm{W}^2_2\left(\mu_c, \hat{\nu}^{(c)}_{\phi_i}\right) - \mathrm{W}^2_2\left(\mu_c, \nu^{(c)}_{\phi_i}\right)\right) \xrightarrow{\text{in distribution}} Z^{(c)}_i,$$

*where $Z^{(c)}_i \sim \mathrm{N}(0, s^{(c)}_i)$ and $s^{(c)}_i = 4(m^{(c)}_i - v_c)^\top \Sigma^{(c)}_i (m^{(c)}_i - v_c) + 2\mathrm{Tr}(\Sigma^{(c)}_i \Sigma_c) - 4\sum_{j=1}^k \kappa_j^{1/2} r_j^\top \Sigma_c^{-1/2} \Sigma^{(c)}_i \Sigma_c^{1/2} r_j$, with $\{\kappa_j, r_j\}_{j \in [k]}$ standing for (eigenvalue, eigenvector) pairs of the symmetric covariance matrix $\Sigma^{(c)}_i$.*

*Proof.* The proof follows from Rippl et al. (2016, Theorem 2.1) with the specific choices $\mu_1 = \nu^{(c)}_{\phi_i}$, $\mu_2 = \mu_c$ and $\hat{\mu}_1 = \hat{\nu}^{(c)}_{\phi_i}$ which are defined in Section 3 in the main paper. $\qquad\square$

### S1.3 Detailed Pseudo-Code for Algorithm 1

In Algorithm S2, we provide algorithmic support to Algorithm 1 in the main paper by detailing how to perform each step. Note that we use the decomposition $\Sigma = LL^\top$ to enforce the positive semi-definite constraint for the covariance matrix $\Sigma$.

---

**Algorithm S2** Detailed version of `FLIC` when using `FedRep`

---

**Require:** initialisation $\alpha^{(0)}$, $\mu_{1:C}^{(0)} = [\Sigma_{1:C}^{(0)}, v_{1:C}^{(0)}]$ with $\Sigma_c^{(0)} = L_c^{(0)}[L_c^{(0)}]^\top$, $\phi_{1:b}^{(0,0)}$, $\beta_{1:b}^{(0,0)}$ and step-size $\eta \leq \bar{\eta}$ for some $\bar{\eta} > 0$.

1: **for** $t = 0$ **to** $T - 1$ **do**
2:    Sample a set of $\mathsf{A}_{t+1}$ of active clients.
3:    **for** $i \in \mathsf{A}_{t+1}$ **do**
4:       The central server sends $\alpha^{(t)}$ and $\mu_{1:C}^{(t)}$ to $\mathsf{A}_{t+1}$.
5:       *// Update local parameters*
6:       **for** $m = 0$ **to** $M - 1$ **do**
7:          Sample a fresh batch $\mathsf{I}_{t+1}^{(i,m)}$ of $n_i'$ samples with $n_i' \in [n_i]$.
8:          Sample $Z_c^{(j,t,m)} \sim \mu_c^{(t)}$ for $j \in \mathsf{I}_{t+1}^{(i,m)}$ and $c \in \mathcal{Y}_i$ via $Z_c^{(j,t,m)} = v_c^{(t)} + L_c^{(t)} \xi^{(t,m)}$ where $\xi_i^{(t,m)} \sim \mathsf{N}(0_k, \mathsf{I}_k)$.

9:          $\phi_i^{(t,m+1)} \quad = \quad \phi_i^{(t,m)} \quad - \quad \eta \frac{n_i}{|\mathsf{I}_{t+1}^{(i,m)}|} \sum_{j \in \mathsf{I}_{t+1}^{(i,m)}} \nabla_{\phi_i} \ell \left( y_i^{(j)}, g_{[\alpha^{(t)}, \beta_i^{(t,m)}]}^{(i)} \left[ \phi_i^{(t,m)} \left( x_i^{(j)} \right) \right] \right) \quad -$

$$\eta \lambda_1 \sum_{c \in \mathcal{Y}_i} \nabla_{\phi_i} \mathsf{W}_2^2 \left( \mu_c^{(t)}, \nu_{\phi_i^{(t,m)}}^{(c)} \right).$$

10:          $\beta_i^{(t,m+1)} \leftarrow \beta_i^{(t,m)} - \eta \frac{n_i}{|\mathsf{I}_{t+1}^{(i,m)}|} \sum_{j \in \mathsf{I}_{t+1}^{(i,m)}} B_j$

11:          with $B_j = \left\{ \nabla_{\beta_i} \ell \left( y_i^{(j)}, g_{[\alpha^{(t)}, \beta_i^{(t,m)}]}^{(i)} \left[ \phi_i^{(t,m)} \left( x_i^{(j)} \right) \right] \right) - \eta \lambda_2 \sum_{c \in \mathcal{Y}_i} \nabla_{\beta_i} \ell \left( y_i^{(j)}, g_{[\alpha^{(t)}, \beta_i^{(t,m)}]}^{(i)} \left[ Z_c^{(j,t,m)} \right] \right) \right\}.$

12:       $\phi_i^{(t+1,0)} = \phi_i^{(t,M)}$.
13:       $\beta_i^{(t+1,0)} = \beta_i^{(t,M)}$.
14:       *// Update global parameters*
15:       $\alpha_i^{(t+1)} \leftarrow \alpha^{(t)} - \eta \frac{n_i}{|\mathsf{I}_{t+1}^{(i,M)}|} \sum_{j \in \mathsf{I}_{t+1}^{(i,M)}} A_j$

16:       with $A_j = \left\{ \nabla_\alpha \ell \left( y_i^{(j)}, g_{[\alpha^{(t)}, \beta_i^{(t,M)}]}^{(i)} \left[ \phi_i^{(t,M)} \left( x_i^{(j)} \right) \right] \right) - \eta \lambda_2 \sum_{c \in \mathcal{Y}_i} \nabla_\alpha \ell \left( y_i^{(j)}, g_{[\alpha^{(t)}, \beta_i^{(t,M)}]}^{(i)} \left[ Z_c^{(j,t,M)} \right] \right) \right\}.$

17:       **for** $c = 1$ **to** $C$ **do**
18:          *// $\hat{m}_i^{(c,t)}, \hat{\Sigma}_i^{(c,t)}$ are the empirical mean and covariance of $\phi_i^{(t,M)} \left( x_i^{(j)} \right)$ for $j \in \mathsf{I}_{t+1}^{(i,M)}$*
19:          *// $v_c^{(t)}$ is the mean of anchor distribution $\mu_c^{(t)}$*
20:          Update $\hat{m}_i^{(c,t)}, \hat{\Sigma}_i^{(c,t)}$ using $\phi_i^{(t,M)}$.
21:          $v_{i,c}^{(t+1)} = v_c^{(t)} - \eta \lambda_1 \nabla_{v_c} \left\| v_c^{(t)} - \hat{m}_i^{(c,t)} \right\|^2 - \eta \lambda_2 \sum_{c \in \mathcal{Y}_i} \frac{n_i}{|\mathsf{I}_{t+1}^{(i,m)}|} \sum_{j \in \mathsf{I}_{t+1}^{(i,m)}} \nabla_{v_c} \ell \left( y_i^{(j)}, g_{[\alpha^{(t)}, \beta_i^{(t,M)}]}^{(i)} \left[ Z_c^{(j,t,M)} \right] \right).$

22:          $L_{i,c}^{(t+1)} = L_c^{(t)} - \eta \lambda_1 \nabla_{L_c} \mathfrak{B}^2 \left( L_c^{(t)} [L_c^{(t)}]^\top, \hat{\Sigma}_i^{(c,t)} \right) - \eta \lambda_2 \sum_{c \in \mathcal{Y}_i} \frac{n_i}{|\mathsf{I}_{t+1}^{(i,m)}|} \sum_{j \in \mathsf{I}_{t+1}^{(i,m)}} \nabla_{L_c} \ell \left( y_i^{(j)}, g_{[\alpha^{(t)}, \beta_i^{(t,M)}]}^{(i)} \left[ Z_c^{(j,t,M)} \right] \right).$

23:       *// Communication with the server*
24:       Send $\alpha_i^{(t+1)}$, $v_{i,1:C}^{(t+1)}$ and $L_{i,1:C}^{(t+1)}$ to central server.
25:    *// Averaging global parameters*
26:    $\alpha^{(t+1)} = \frac{b}{|\mathsf{A}_{t+1}|} \sum_{i \in \mathsf{A}_{t+1}} w_i \alpha_i^{(t+1)}$.
27:    **for** $c = 1$ **to** $C$ **do**
28:       $v_c^{(t+1)} = (b/|\mathsf{A}_{t+1}|) \sum_{i \in \mathsf{A}_{t+1}} \omega_i v_{i,c}^{(t+1)}$.
29:       $L_c^{(t+1)} = (b/|\mathsf{A}_{t+1}|) \sum_{i \in \mathsf{A}_{t+1}} \omega_i L_{i,c}^{(t+1)}$ and set $\Sigma_c^{(t+1)} = L_c^{(t+1)} [L_c^{(t+1)}]^\top$.

**Ensure:** parameters $\alpha^{(T)}$, $\mu_{1:C}^{(T)}$, $\phi_{1:b}^{(T,0)}$, $\beta_{1:b}^{(T,0)}$.

---

## S1.4   Additional Algorithmic Insights

**Scalability.** When the number of classes $C$ is large, both local computation and communication costs are increased. In this setting, we propose to partition all the classes into $C_{\text{meta}} \ll C$ meta-classes and consider reference measures $\{\mu_c\}_{c \in [C_{\text{meta}}]}$ associated to these meta-classes. As an example, if we are considering a dataset made of features associated to animals, the meta-class refers to an animal (*e.g.* a dog) and the class refers to a specific breed (*e.g.* golden retriever).

**Privacy Consideration.** As other standard (personalised) FL algorithms, `FLIC` satisfies first-order privacy guarantees by not allowing raw data exchanges but rather exchanges of local Gaussian statistics. Note that `FLIC` stands for a post-hoc approach and can be combined with other privacy/confidentiality techniques such as differential privacy (Dwork & Roth, 2014), secure aggregation via secure multi-party computation (Chen et al., 2022) or trusted execution environments (Mo et al., 2021).

**Inference on New Clients.** When a client who has not participated to the training procedure appears, there is no need to re-launch a potentially costly federated learning procedure. Instead, the server sends the shared parameters $\{\alpha^{(T)}, \mu_{1:C}^{(T)}\}$ to the new client and the latter only needs to learn the local parameters $\{\phi_i, \beta_i\}$.

## S2 Proof of Theorem 1

This section aims at proving Theorem 1 in the main paper. To this end, we first provide in Appendix S2.1 a closed-form expression for the estimated embedded features based on the features embedded by the oracle. Then, in Appendix S2.3, we show technical lemmata that will be used in Appendix S2.2 to show Theorem 1.

To prove our results, we consider the following set of assumptions.

**H1.** *(i) For any $i \in [b]$, $j \in [n_i]$, ground-truth embedded features $\phi_i^\star(x_i^{(j)})$ are distributed according to $N(0_k, I_k)$.*
*(ii) Ground-truth model parameters satisfy $\|\beta_i^\star\|_2 = \sqrt{d}$ for $i \in [b]$ and $A^\star$ has orthonormal columns.*
*(iii) For any $t \in \{0, \ldots, T-1\}, |A_{t+1}| = \lfloor rb \rfloor$ with $1 \le \lfloor rb \rfloor \le b$, and if we select $\lfloor rb \rfloor$ clients, their ground-truth head parameters $\{\beta_i^\star\}_{i \in A_{t+1}}$ span $\mathbb{R}^d$.*
*(iv) In (2) in the main paper, $\ell(\cdot, \cdot)$ is the $\ell_2$ norm, $\omega_i = 1/b$, $\theta_i = [A, \beta_i]$ and $g_{\theta_i}^{(i)}(x) = (A\beta_i)^\top x$ for $x \in \mathbb{R}^k$.*

### S2.1 Estimation of the Feature Transformation Functions

As in Section 5 in the main paper, we assume that $x_i^{(j)} \sim N(m_i, \Sigma_i)$ with $m_i \in \mathbb{R}^{k_i}$ and $\Sigma_i \in \mathbb{R}^{k_i \times k_i}$ for $i \in [b], j \in [n_i]$. In addition, we consider that the continuous scalar labels are generated via the oracle model $y_i^{(j)} = (A^\star \beta_i^\star)^\top \phi_i^\star(x_i^{(j)})$ where $A^\star \in \mathbb{R}^{k \times d}$, $\beta_i^\star \in \mathbb{R}^d$ and $\phi_i^\star(\cdot)$ are ground-truth parameters and feature transformation function, respectively. Under **H**1-(i), the oracle feature transformation functions $\{\phi_i^\star\}_{i \in [b]}$ are assumed to map $k_i$-dimensional Gaussian distributions $N(m_i, \Sigma_i)$ to a common $k$-dimension Gaussian $N(0_k, I_k)$. As shown in Delon et al. (2022, Theorem 4.1), there exist closed-form expressions for $\{\phi_i^\star\}_{i \in [b]}$, which can be shown to stand for solutions of a Gromov-Wasserstein problem restricted to Gaussian transport plans. More precisely, these oracle feature transformation stand for affine maps and are of the form, for any $i \in [b]$,

$$\phi_i^\star\left(x_i^{(j)}\right) = \left[\tilde{I}_k^{(i,\star)}(D_i^{(k)})^{-1/2}P_i^\top \quad 0_{k,k_i-k}\right]\left(x_i^{(j)} - m_i\right),$$

where $\tilde{I}_k^{(i,\star)} = \mathrm{diag}_k(\pm 1)$ is a $k$-dimensional diagonal matrix with diagonal elements in $\{-1, 1\}$, $\Sigma_i = P_i D_i P_i^\top$ is the diagonalisation of $\Sigma_i$ and $D_i^{(k)}$ stands for the restriction of $D_i$ to the first $k$ components. In the sequel, we assume that all oracle feature transformation functions share the same randomness, that is $\tilde{I}_k^{(i,\star)} = \tilde{I}_k^\star = \mathrm{diag}_k(\pm 1)$.

For the sake of simplicity, we assume that we know the true latent distribution of $\phi_i^\star(x_i^{(j)})$ and as such consider a pre-fixed reference latent distribution that equals the latter, that is $\mu = N(0_k, I_k)$. Since we know from Delon et al. (2022, Theorem 4.1) that there exist mappings between Gaussian distributions with supports associated to different metric spaces, we propose an estimate for the ground-truth feature transformation functions defined by for any $i \in [b]$,

$$\hat{\phi}_i\left(x_i^{(j)}\right) = \left[\tilde{I}_k(D_i^{(k)})^{-1/2}P_i^\top \quad 0_{k,k_i-k}\right]\left(x_i^{(j)} - m_i\right),$$

where $\tilde{I}_k = \mathrm{diag}_k(\pm 1)$. By noting that $\tilde{I}_k = Q\tilde{I}_k^\star$, where $Q \in \mathbb{R}^{k \times k}$ is a diagonal matrix of the form $\mathrm{diag}_k(\pm 1)$, it follows that

$$\hat{\phi}_i\left(x_i^{(j)}\right) = Q\phi_i^\star\left(x_i^{(j)}\right). \tag{S1}$$

---

**Algorithm S3** `FLIC-FedRep` for linear regression and Gaussian features

---

**Require:** step size $\eta$, number of outer iterations $T$, participation rate $r \in (0,1)$, diagonalizations $\Sigma_i = P_i D_i P_i^\top$ sorting eigenvalues in decreasing order.

1: *// Estimation of embedded features*

2: For each client $i \in [b]$, set $\hat{\phi}_i \left( x_i^{(j)} \right) = \begin{bmatrix} \tilde{I}_k (D_i^{(k)})^{-1/2} P_i^\top & 0_{k, k_i - k} \end{bmatrix} \left( x_i^{(j)} - m_i \right)$.

3: *// Initialisation $A^{(0)}$*

4: Each client $i \in [b]$ sends $Z_i = (1/n_i) \sum_{j=1}^{n_i} (y_i^{(j)})^2 \hat{\phi}_i \left( x_i^{(j)} \right) [\hat{\phi}_i \left( x_i^{(j)} \right)]^\top$ to the central server.

5: The central server computes $UDU^\top \leftarrow \text{rank}-d \ \text{SVD} \left( (1/b) \sum_{i=1}^b Z_i \right)$.

6: The central server initialises $A^{(0)} = U$.

7: **for** $t = 0$ **to** $T - 1$ **do**

8:     Sample a set of $\mathsf{A}_{t+1}$ of active clients such that $|\mathsf{A}_{t+1}| = \lfloor rb \rfloor$.

9:     **for** $i \in \mathsf{A}_{t+1}$ **do**

10:        The central server sends $A^{(t)}$ to $\mathsf{A}_{t+1}$.

11:        *// Update local parameters*

12:        $\beta_i^{(t+1)} = \arg\min_{\beta_i} \sum_{j=1}^{n_i} \left( y_i^{(j)} - \beta_i^\top [A^{(t)}]^\top \hat{\phi}_i \left( x_i^{(j)} \right) \right)^2$.

13:        *// Update global parameters*

14:        $A_i^{(t+1)} = A^{(t)} - \eta \nabla_A \sum_{j=1}^{n_i} \left( y_i^{(j)} - [\beta_i^{(t+1)}]^\top A^\top \hat{\phi}_i \left( x_i^{(j)} \right) \right)^2$.

15:        *// Communication with the server*

16:        Send $A_i^{(t+1)}$ to the central server.

17:     *// Averaging and orthogonalisation of global parameter*

18:     $\bar{A}^{(t+1)} = \frac{1}{\lfloor rb \rfloor} \sum_{i \in \mathsf{A}_{t+1}} A_i^{(t+1)}$.

19:     $A^{(t+1)}, R^{(t+1)} \leftarrow \text{QR} \left( \bar{A}^{(t+1)} \right)$.

**Ensure:** parameters $A^{(T)}$, $\beta_{1:b}^{(T)}$.

---

In Appendix S2.2, the equation (S1) will allow us to relate the ground-truth labels $y_i^{(j)} = (A^\star \beta_i^\star)^\top \phi_i^\star (x_i^{(j)})$ with estimated predictions $\hat{y}_i^{(j)} = (A^{(T)} \beta_i^{(T)})^\top \hat{\phi}_i (x_i^{(j)})$ via Algorithm S3 starting from the same embedded features.

## S2.2   Proof of Theorem 1

Let $B \in \mathbb{R}^{b \times d}$ the matrix having local model parameters $\{\beta_i\}_{i \in [b]}$ as columns and denote by $B_{\mathsf{A}_{t+1}} \in \mathbb{R}^{\lfloor rb \rfloor \times d}$ its restriction to the row set defined by $\mathsf{A}_{t+1}$ where $|\mathsf{A}_{t+1}| = \lfloor rb \rfloor$ for some $r \in (0,1]$. For the sake of simplicity, we assume in the sequel that all clients have the same number of data points that is for any $i \in [b]$, $n_i = n$. For random batches of samples $\{(x_i^{(j)}, y_i^{(j)}), j \in [n]\}_{i \in [\lfloor rb \rfloor]}$, we define similarly to Collins et al. (2021); Jain et al. (2013), the random linear operator $\mathcal{A} : \mathbb{R}^{\lfloor rb \rfloor \times d} \to \mathbb{R}^{\lfloor rb \rfloor n}$ for any $M \in \mathbb{R}^{\lfloor rb \rfloor \times d}$ as $\mathcal{A}(M) = [\langle e_i (\phi_i^\star (x_i^{(j)}))^\top, M \rangle]_{1 \le i \le \lfloor rb \rfloor, 1 \le j \le n}$, where $e_i$ stands for the $i$-th standard vector of $\mathbb{R}^{\lfloor rb \rfloor}$. Using these notations, it follows from Algorithm S3 that for any $t \in \{0, \ldots, T-1\}$, the model parameters $\theta_i^{(t+1)} = [A^{(t+1)}, \beta_i^{(t+1)}]$ are computed as follows:

$$B_{\mathsf{A}_{t+1}}^{(t+1)} = \arg\min_{B_{\mathsf{A}_{t+1}}} \frac{1}{\lfloor rb \rfloor \, n} \left\| \mathcal{A}^{(t+1)} \left( B_{\mathsf{A}_{t+1}}^\star [A^\star]^\top - B_{\mathsf{A}_{t+1}} [A^{(t)}]^\top Q \right) \right\|^2, \tag{S2}$$

$$\bar{A}^{(t+1)} = \bar{A}^{(t)} - \frac{\eta}{\lfloor rb \rfloor \, n} \left[ (\mathcal{A}^{(t+1)})^\dagger \mathcal{A}^{(t+1)} \left( B_{\mathsf{A}_{t+1}}^\star [A^\star]^\top - B_{\mathsf{A}_{t+1}}^{(t+1)} [A^{(t)}]^\top Q \right) \right]^\top Q B_{\mathsf{A}_{t+1}}^{(t+1)},$$

$$A^{(t+1)}, R^{(t+1)} \leftarrow \text{QR} \left( \bar{A}^{(t+1)} \right), \tag{S3}$$

where $\mathcal{A}^{(t+1)}$ stands for a specific instance of $\mathcal{A}$ depending on the random subset of active clients available at each round and $\mathcal{A}^{\dagger}$ is the adjoint operator of $\mathcal{A}$ defined by $\mathcal{A}^{\dagger}(M) = \sum_{i \in [\lfloor rb \rfloor]} \sum_{i=1}^{n} [\langle e_i(\phi_i^{\star}(x_i^{(j)}))^{\top}, M \rangle] e_i(\phi_i^{\star}(x_i^{(j)}))$.

The update in (S2) admits a closed-form expression as shown in the following lemma.

**Lemma S1.** *For any $t \in \ldots 0, \ldots, T-1$, we have*

$$B_{\mathsf{A}_{t+1}}^{(t+1)} = B_{\mathsf{A}_{t+1}}^{\star}[A^{\star}]^{\top} Q A^{(t)} - F^{(t)},$$

*where $F^{(t)}$ is defined in (S12), $A^{(t)}$ is defined in (S3) and $B_{\mathsf{A}_t}^{(t)}$ is defined in (S2).*

*Proof.* The proof follows from the same steps as in Collins et al. (2021, Proof of Lemma 1) using (S2). $\square$

Under **H**1, we have the following non-asymptotic convergence result.

**Theorem S3.** *Assume **H**1. Then, for any $x_i \in \mathbb{R}^{k_i}$, we have $\hat{\phi}_i(x_i) = Q\phi_i^{\star}(x_i)$ where $Q \in \mathbb{R}^{k \times k}$ is of the form $\operatorname{diag}_k(\pm 1)$. Define $E_0 = \operatorname{dist}(A^{(0)}, QA^{\star})$. Assume that $n \geq c(d^3 \log(\lfloor rb \rfloor))/E_0^2 + d^2 k/(E_0^2 \lfloor rb \rfloor)$ for some absolute constant $c > 0$. Then, for any $t \in \{0, \ldots, T-1\}$, $\eta \leq 1/(4\bar{\sigma}_{\max,\star}^2)$ and with high probability at least $1 - e^{-110k} - e^{-110d^2 \log(\lfloor rb \rfloor)}$, we have*

$$\operatorname{dist}(A^{(t+1)}, QA^{\star}) \leq (1-\kappa)^{(t+1)/2} \operatorname{dist}(A^{(0)}, QA^{\star}),$$

*where $A^{(t)}$ is computed via Algorithm S3, $\operatorname{dist}$ denotes the principal angle distance and $\kappa \in (0,1)$ is defined as*

$$\kappa = 1 - \eta E_0 \bar{\sigma}_{\min,\star}^2/2.$$

*Proof.* The proof follows first by plugging Lemma S3, Lemma S8 and Lemma S9 into Lemma S2. Then, we use the same technical arguments and steps as in Collins et al. (2021, Proof of Lemma 6). $\square$

## S2.3 Technical Lemmata

In this section, we provide a set of useful technical lemmata to prove our main result in Appendix S2.2.

**Notations.** We begin by defining some notations that will be used in the sequel. For any $t \in \{0, \ldots, T-1\}$, we define

$$Z^{(t+1)} = B_{\mathsf{A}_{t+1}}^{(t+1)}[A^{(t)}]^{\top} Q - B_{\mathsf{A}_{t+1}}^{\star}[A^{\star}]^{\top}. \tag{S4}$$

In addition, let

$$G^{(t)} = \begin{bmatrix} G_{11}^{(t)} & \cdots & G_{1d}^{(t)} \\ \vdots & \ddots & \vdots \\ G_{d1}^{(t)} & \cdots & G_{dd}^{(t)} \end{bmatrix}, C^{(t)} = \begin{bmatrix} C_{11}^{(t)} & \cdots & C_{1d}^{(t)} \\ \vdots & \ddots & \vdots \\ C_{d1}^{(t)} & \cdots & C_{dd}^{(t)} \end{bmatrix}, D^{(t)} = \begin{bmatrix} D_{11}^{(t)} & \cdots & D_{1d}^{(t)} \\ \vdots & \ddots & \vdots \\ D_{d1}^{(t)} & \cdots & D_{dd}^{(t)} \end{bmatrix},$$

where for $p, q \in [d]$,

$$G_{pq}^{(t)} = \frac{1}{n} \sum_{i \in \mathsf{A}_{t+1}} \sum_{j=1}^{n} e_i \left(\phi_i^{\star}(x_i^{(j)})\right)^{\top} Q a_p^{(t)} [a_q^{(t)}]^{\top} Q \phi_i^{\star}(x_i^{(j)}) e_i^{\top}, \tag{S5}$$

$$C_{pq}^{(t)} = \frac{1}{n} \sum_{i \in \mathsf{A}_{t+1}} \sum_{j=1}^{n} e_i \left(\phi_i^{\star}(x_i^{(j)})\right)^{\top} Q a_p^{(t)} [a_q^{\star}]^{\top} Q \phi_i^{\star}(x_i^{(j)}) e_i^{\top}, \tag{S6}$$

$$D_{pq}^{(t)} = \langle a_p^{(t)}, a_q^{\star} \rangle \mathrm{I}_{\lfloor rb \rfloor}, \tag{S7}$$

with $a_p^{(t)} \in \mathbb{R}^k$ standing for the $p$-th column of $A^{(t)} \in \mathbb{R}^{k \times d}$; and $a_p^\star \in \mathbb{R}^k$ standing for the $p$-th column of $A^\star \in \mathbb{R}^{k \times d}$. Finally, we define for any $i \in \mathsf{A}_{t+1}$,

$$\Pi^i = \frac{1}{n} \sum_{j=1}^{n} \phi_i^\star(x_i^{(j)})[\phi_i^\star(x_i^{(j)})]^\top , \tag{S8}$$

$$(G^{(t)})^i = [A^{(t)}]^\top Q \Pi^i Q A^{(t)} , \tag{S9}$$

$$(C^{(t)})^i = [A^{(t)}]^\top Q \Pi^i Q A^\star , \tag{S10}$$

$$(D^{(t)})^i = [A^{(t)}]^\top Q A^\star . \tag{S11}$$

Using these notations, we also define $\tilde{\beta}^\star = [(\beta_1^\star)^\top, \ldots, (\beta_d^\star)^\top]^\top \in \mathbb{R}^{\lfloor rb \rfloor d}$ and

$$F^{(t)} = [([G^{(t)}]^{-1}(G^{(t)}D^{(t)} - C^{(t)})\tilde{\beta}^\star)_1, \ldots, ([G^{(t)}]^{-1}(G^{(t)}D^{(t)} - C^{(t)})\tilde{\beta}^\star)_d] . \tag{S12}$$

**Technical results.** To prove our main result in Theorem S3, we begin by providing a first upper bound on the quantity of interest namely dist $\left(A^{(t+1)}, QA^\star\right)$. This is the purpose of the next lemma.

**Lemma S2.** *For any $t \in \{0, \ldots, T-1\}$ and $\eta > 0$, we have*

$$\mathrm{dist}\left(A^{(t+1)}, QA^\star\right) \le C_1 + C_2, ,$$

*where*

$$C_1 = \left\| [A_\perp^\star]^\top Q A^{(t)} \left( \mathrm{I}_d - \frac{\eta}{\lfloor rb \rfloor} [B_{\mathsf{A}_{t+1}}^{(t+1)}]^\top B_{\mathsf{A}_{t+1}}^{(t+1)} \right) \right\|_2 \left\| \left(R^{(t+1)}\right)^{-1} \right\|_2 , \tag{S13}$$

$$C_2 = \frac{\eta}{\lfloor rb \rfloor} \left\| \left( \frac{1}{n} [A_\perp^\star]^\top (Q\mathcal{A}^{(t+1)})^\dagger \mathcal{A}^{(t+1)} \left(Z^{(t+1)}\right) Q - Z^{(t+1)} \right)^\top B_{\mathsf{A}_{t+1}}^{(t+1)} \right\|_2 \left\| \left(R^{(t+1)}\right)^{-1} \right\|_2 , \tag{S14}$$

*where $A^{(t)}$ is defined in (S3), $B_{\mathsf{A}_t}^{(t)}$ is defined in (S2), $Z^{(t)}$ is defined in (S4) and $R^{(t)}$ comes from the QR factorisation of $\bar{A}^{(t)}$, see step 19 in Algorithm S3.*

*Proof.* The proof follows from the same steps as in Collins et al. (2021, Proof of Lemma 6) and by noting that dist$(A^{(t)}, QA^\star) = $ dist$(QA^{(t)}, A^\star)$ for $t \in \{0, \ldots, T-1\}$. $\square$

We now have to control the terms $C_1$ and $C_2$. For the sake of clarity, we split technical results aiming to upper bound of $C_1$ and $C_2$ in two different paragraphs.

**Control of $C_1$.**

**Lemma S3.** *Assume **H**1. Let $\delta_d = cd^{3/2}\sqrt{\log(\lfloor rb \rfloor)}/n^{1/2}$ for some absolute constant $c > 0$. Then, for any $t \in \{0, \ldots, T-1\}$, with probability at least $1 - e^{-111k^2 \log(\lfloor rb \rfloor)}$, we have for $\delta_d \le 1/2$ and $\eta \le 1/(4\bar{\sigma}_{\max,\star}^2)$*

$$C_1 \le \left[ \le 1 - \eta \left(1 - \mathrm{dist}\left(A^{(0)}, QA^\star\right)\right) \bar{\sigma}_{\min,\star}^2 + 2\eta \frac{\delta_d}{1-\delta_d} \bar{\sigma}_{\max}^2 \right] \mathrm{dist}\left(A^{(t)}, QA^\star\right) \left\| \left(R^{(t+1)}\right)^{-1} \right\|_2 ,$$

*where $\bar{\sigma}_{\min}^2, \bar{\sigma}_{\max}^2$ are defined in (S15)-(S16), $C_1$ is defined in (S13), $A^{(t)}$ is defined in (S3) and $R^{(t)}$ comes from the QR factorisation of $\bar{A}^{(t)}$, see step 19 in Algorithm S3.*

*Proof.* Using Cauchy-Schwarz inequality, we have

$$C_1 \le \left\| (A_\perp^\star)^\top Q A^{(t)} \right\|_2 \left\| \mathrm{I}_d - \frac{\eta}{\lfloor rb \rfloor} [B_{\mathsf{A}_{t+1}}^{(t+1)}]^\top B_{\mathsf{A}_{t+1}}^{(t+1)} \right\|_2 \left\| \left(R^{(t+1)}\right)^{-1} \right\|_2$$

$$= \mathrm{dist}\left(A^{(t)}, QA^\star\right) \left\| \mathrm{I}_d - \frac{\eta}{\lfloor rb \rfloor} [B_{\mathsf{A}_{t+1}}^{(t+1)}]^\top B_{\mathsf{A}_{t+1}}^{(t+1)} \right\|_2 \left\| \left(R^{(t+1)}\right)^{-1} \right\|_2 .$$

Define the following minimum and maximum singular values:

$$\bar{\sigma}_{\min,\star}^2 = \min_{\mathsf{A} \subseteq [b], |\mathsf{A}| = \lfloor rb \rfloor} \sigma_{\min} \left( \frac{1}{\sqrt{\lfloor rb \rfloor}} B_{\mathsf{A}}^\star \right) \tag{S15}$$

$$\bar{\sigma}_{\max,\star}^2 = \min_{\mathsf{A} \subseteq [b], |\mathsf{A}| = \lfloor rb \rfloor} \sigma_{\max} \left( \frac{1}{\sqrt{\lfloor rb \rfloor}} B_{\mathsf{A}}^\star \right) . \tag{S16}$$

Using Collins et al. (2021, Proof of Lemma 6, equations (67)-(68)), we have for $\delta_d \leq 1/2$ where $\delta_d$ is defined in Lemma S4 and $\eta \leq 1/(4\bar{\sigma}_{\max,\star}^2)$,

$$\left\| \mathrm{I}_d - \frac{\eta}{\lfloor rb \rfloor} [B_{\mathsf{A}_{t+1}}^{(t+1)}]^\top B_{\mathsf{A}_{t+1}}^{(t+1)} \right\|_2 \leq 1 - \eta \left( 1 - \mathrm{dist} \left( A^{(0)}, QA^\star \right) \right) \bar{\sigma}_{\min,\star}^2 + 2\eta \frac{\delta_d}{1 - \delta_d} \bar{\sigma}_{\max,\star}^2 ,$$

with probability at least $1 - e^{-111k^2 \log(\lfloor rb \rfloor)}$ The proof is concluded by combining the two previous bounds. $\quad\square$

**Control of $C_2$.** We begin by showing four intermediary results gathered in the next four lemmata.

**Lemma S4.** *Assume **H**1. Let $\delta_d = cd^{3/2}\sqrt{\log(\lfloor rb \rfloor)}/n^{1/2}$ for some absolute constant $c > 0$. Then, for any $t \in \{0, \ldots, T-1\}$, with probability at least $1 - e^{-111k^3 \log(\lfloor rb \rfloor)}$, we have*

$$\left\| [G^{(t)}]^{-1} \right\|_2 \leq \frac{1}{1 - \delta_d} ,$$

*where $G^{(t)}$ is defined in (S5).*

*Proof.* The proof stands as a straightforward extension of Collins et al. (2021, Proof of Lemma 2) by noting that the random variable $Q\phi_i^\star(x_i^{(j)}) = \hat{\phi}_i(x_i^{(j)})$ is sub-Gaussian under **H**1-(i); and as such is omitted. $\quad\square$

**Lemma S5.** *Assume **H**1. Let $\delta_d = cd^{3/2}\sqrt{\log(\lfloor rb \rfloor)}/n^{1/2}$ for some absolute constant $c > 0$. Then, for any $t \in \{0, \ldots, T-1\}$, with probability at least $1 - e^{-111k^2 \log(\lfloor rb \rfloor)}$, we have*

$$\left\| (G^{(t)} D^{(t)} - C^{(t)}) B_{\mathsf{A}_t}^\star \right\|_2 \leq \delta_d \left\| B_{\mathsf{A}_t}^\star \right\|_2 \mathrm{dist} \left( A^{(t)}, QA^\star \right) ,$$

*where $G^{(t)}$ is defined in (S5), $D^{(t)}$ is defined in (S7), $C^{(t)}$ is defined in (S6) and $A^{(t)}$ in (S3).*

*Proof.* Without loss of generality and to ease notation, we remove the superscript $(t)$ in the proof and re-index the indexes of clients in $\mathsf{A}_{t+1}$. Let $H = GD - C$. From (S8), (S9), (S10) and (S11), it follows, for any $i \in [\lfloor rb \rfloor]$, that

$$H^i = G^i D^i - C^i = A^\top Q \Pi^i Q (AA^\top - \mathrm{I}_k) QA^\star .$$

Hence, by using the definition of $H$, we have

$$\| (GD - C)\beta^\star \|_2^2 = \sum_{i=1}^{\lfloor rb \rfloor} \left\| H^i \beta_i^\star \right\|_2^2 \leq \sum_{i=1}^{\lfloor rb \rfloor} \left\| H^i \right\|_2^2 \|\beta_i^\star\|^2 \leq \frac{d}{\lfloor rb \rfloor} \| B^\star \|_2^2 \sum_{i=1}^{\lfloor rb \rfloor} \left\| H^i \right\|_2^2 ,$$

where the last inequality follows almost surely from **H**1-(iii). As in Collins et al. (2021, Proof of Lemma 3), we then define for any $j \in [n]$, the vectors

$$u_i^{(j)} = \frac{1}{\sqrt{n}} [A^\star]^\top (AA^\top - \mathrm{I}_k) Q\phi_i^\star(x_i^{(j)}) ,$$

$$v_i^{(j)} = \frac{1}{\sqrt{n}} A^\top Q\phi_i^\star(x_i^{(j)}) .$$

Let $\mathcal{S}^{d-1}$ denotes the $d$-dimensional unit spheres. Then, by Vershynin (2018, Corollary 4.2.13), we can define $\mathcal{N}_d$, the $1/4$-net over $\mathcal{S}^{d-1}$ such that $|\mathcal{N}_d| \leq 9^d$. Therefore, by using Vershynin (2018, Equation (4.13)), we have

$$\left\| H^i \right\|_2^2 \leq 2 \max_{z,y \in \mathcal{N}_d} \sum_{j=1}^{n} \langle z, u_i^{(j)} \rangle \langle v_i^{(j)}, y \rangle \,.$$

Since $\phi_i^\star(x_i^{(j)})$ is a standard Gaussian vector, it is sub-Gaussian and therefore $\langle z, u_i^{(j)} \rangle$ and $\langle v_i^{(j)}, y \rangle$ are sub-Gaussian with norms $\| \frac{1}{\sqrt{n}} [A^\star]^\top (AA^\top - \mathrm{I}_k) Q \|_2 = (1/\sqrt{n}) \mathrm{dist}(A, QA^\star)$ and $(1/\sqrt{n})$, respectively. In addition, we have

$$\begin{aligned}
\mathbb{E}\left[ \langle z, u_i^{(j)} \rangle \langle v_i^{(j)}, y \rangle \right] &= \frac{1}{n} \mathbb{E}\left[ z^\top \frac{1}{\sqrt{n}} [A^\star]^\top (AA^\top - \mathrm{I}_k) Q \phi_i^\star(x_i^{(j)}) [\phi_i^\star(x_i^{(j)})]^\top Q A y \right] \\
&= \frac{1}{n} z^\top \frac{1}{\sqrt{n}} [A^\star]^\top (AA^\top - \mathrm{I}_k) A y \\
&= 0,
\end{aligned}$$

where we have used the fact that $\mathbb{E}[\phi_i^\star(x_i^{(j)})[\phi_i^\star(x_i^{(j)})]^\top] = 1$, $Q^2 = \mathrm{I}_k$ and $(AA^\top - \mathrm{I}_k)A = 0$. The rest of the proof is concluded by using the Bernstein inequality by following directly the steps detailed in Collins et al. (2021, Proof of Lemma 3, see equations (35) to (39)). $\qquad\square$

**Lemma S6.** *Assume **H1**. Let $\delta_d = cd^{3/2}\sqrt{\log(\lfloor rb \rfloor)}/n^{1/2}$ for some absolute constant $c > 0$. Then, for any $t \in [T]$, with probability at least $1 - e^{-111k^2 \log(\lfloor rb \rfloor)}$, we have*

$$\left\| F^{(t)} \right\|_F \leq \frac{\delta_d}{1 - \delta_d} \left\| B_{\mathsf{A}_t}^\star \right\|_2 \mathrm{dist}\left( A^{(t)}, QA^\star \right),$$

*where $F^{(t)}$ is defined in (S12) and $A^{(t)}$ in (S3).*

*Proof.* By the Cauchy-Schwarz inequality, we have $\left\| F^{(t)} \right\|_F = \left\| [G^{(t)}]^{-1} (G^{(t)} D^{(t)} - C^{(t)}) B_{\mathsf{A}_t}^\star \right\|_2 \leq \delta_d \left\| B_{\mathsf{A}_t}^\star \right\|_2 \leq \left\| [G^{(t)}]^{-1} \right\|_2 \left\| (G^{(t)} D^{(t)} - C^{(t)}) B_{\mathsf{A}_t}^\star \right\|_2 \leq \delta_d \left\| B_{\mathsf{A}_t}^\star \right\|_2$. The proof is concluded by combining the upper bounds given in Lemma S4 and Lemma S5. $\qquad\square$

**Lemma S7.** *Assume **H1** and let $\delta_d' = cd\sqrt{k}/\sqrt{\lfloor rb \rfloor n}$ for some absolute positive constant $c$. For any $t \in [T]$ and whenever $\delta_d' \leq d$, we have with probability at least $1 - e^{-110k} - e^{-110d^2 \log(\lfloor rb \rfloor)}$*

$$\frac{1}{\lfloor rb \rfloor} \left\| \left( \frac{1}{n} Q (\mathcal{A}^{(t)})^\dagger \mathcal{A}^{(t)} \left( Z^{(t)} \right) Q - Z^{(t)} \right)^\top B_{\mathsf{A}_t}^{(t)} \right\|_2 \leq \delta_d' \, \mathrm{dist}\left( A^{(t)}, QA^\star \right),$$

*where $B_{\mathsf{A}_t}^{(t)}$ is defined in (S2) and $Z^{(t)}$ is defined in (S4).*

*Proof.* Let $t \in [T]$. Note that we have

$$\left( \frac{1}{n} Q (\mathcal{A}^{(t)})^\dagger \mathcal{A}^{(t)} \left( Z^{(t)} \right) Q - Z^{(t)} \right)^\top B_{\mathsf{A}_t}^{(t)} = \frac{1}{n} \sum_{i \in \mathsf{A}_t} \sum_{j=1}^{m} \langle Q \phi_i^\star(x_i^{(j)}), z_i^{(t)} \rangle Q \phi_i^\star(x_i^{(j)}) \left[ \beta_i^{(t)} \right]^\top - z_i^{(t)} \left[ \beta_i^{(t)} \right]^\top.$$

Let $\mathcal{S}^{k-1}$ and $\mathcal{S}^{d-1}$ denote the $k$-dimensional and $d$-dimensional unit spheres, respectively. Then, by Vershynin (2018, Corollary 4.2.13), we can define $\mathcal{N}_k$ and $\mathcal{N}_d$, $1/4$-nets over $\mathcal{S}^{k-1}$ and $\mathcal{S}^{d-1}$, respectively, such that $|\mathcal{N}_k| \leq 9^k$ and $|\mathcal{N}_d| \leq 9^d$. Therefore, by using Vershynin (2018, Equation (4.13)), we have

$$\left\| \left( \frac{1}{n} Q (\mathcal{A}^{(t)})^\dagger \mathcal{A}^{(t)} \left( Z^{(t)} \right) Q - Z^{(t)} \right)^\top B_{\mathsf{A}_t}^{(t)} \right\|_2^2$$

$$= 2 \max_{u \in \mathcal{N}_d, v \in \mathcal{N}_k} u^\top \left[ \frac{1}{n} \sum_{i \in A_t} \sum_{j=1}^m \langle Q\phi_i^\star(x_i^{(j)}), z_i^{(t)} \rangle Q\phi_i^\star(x_i^{(j)}) \left[ \beta_i^{(t)} \right]^\top - z_i^{(t)} \left[ \beta_i^{(t)} \right]^\top \right] v$$

$$= 2 \max_{u \in \mathcal{N}_d, v \in \mathcal{N}_k} \frac{1}{n} \sum_{i \in A_t} \sum_{j=1}^m \langle Q\phi_i^\star(x_i^{(j)}), z_i^{(t)} \rangle \langle u, Q\phi_i^\star(x_i^{(j)}) \rangle \langle \beta_i^{(t)}, v \rangle - \langle u, z_i^{(t)} \rangle \langle \beta_i^{(t)}, v \rangle. \qquad \text{(S17)}$$

In order to control (S17) using Bernstein inequality as in Lemma S5, we need to characterise, in particular, the sub-Gaussianity of $\langle u, z_i^{(t)} \rangle$ and $\langle \beta_i^{(t)}, v \rangle$ which require a bound on $\|z_i^{(t)}\|$ and $\|\beta_i^{(t)}\|$, respectively. From Lemma S1, we have $[\beta_i^{(t)}]^\top = (\beta_i^\star)^\top (A^\star)^\top A^{(t)} - (z_i^{(t)})^\top$ which leads to

$$\left\| z_i^{(t)} \right\|^2 = \left\| QA^{(t)}(A^{(t)})^\top QA^\star \beta_i^\star - QA^{(t)} f_i^{(t)} - A^\star \beta_i^\star \right\|_2^2$$

$$= \left\| (QA^{(t)}(A^{(t)})^\top Q - I_d)A^\star \beta_i^\star - QA^{(t)} f_i^{(t)} \right\|_2^2$$

$$\leq 2 \left\| (QA^{(t)}(A^{(t)})^\top Q - I_d)A^\star \right\|_2^2 \|\beta_i^\star\|^2 + 2 \left\| f_i^{(t)} \right\|^2$$

$$\leq 2d \operatorname{dist}^2(A^{(t)}, QA^\star) + 2 \left\| f_i^{(t)} \right\|^2.$$

Using (S12) and the Cauchy-Schwarz inequality, we have

$$\left\| f_i^{(t)} \right\|^2 = \left\| [G^{i,(t)}]^{-1}(G^{i,(t)} D^{i,(t)} - C^{i,(t)})\beta_i^\star \right\|^2$$

$$\leq \left\| [G^{i,(t)}]^{-1} \right\|_2^2 \left\| G^{i,(t)} D^{i,(t)} - C^{i,(t)} \right\|_2^2 \|\beta_i^\star\|^2$$

$$\leq d \left\| [G^{i,(t)}]^{-1} \right\|_2^2 \left\| G^{i,(t)} D^{i,(t)} - C^{i,(t)} \right\|_2^2, \qquad \text{(S18)}$$

where the last inequality follows from **H**1-(ii).

Using Lemma S4 and Lemma S5 and similarly to Collins et al. (2021, Equation (45)), it follows for any $i \in A_t$ that

$$\left\| z_i^{(t)} \right\|_2^2 \leq 4d \operatorname{dist}(A^{(t)}, QA^\star),$$

with probability at least $1 - e^{110d^2 \log(\lfloor rb \rfloor)}$.

Similarly, using Lemma S1 and (S18), we have with probability at least $1 - e^{110d^2 \log(\lfloor rb \rfloor)}$ and for any $i \in A_t$, that

$$\left\| \beta_i^{(t)} \right\|^2 \leq 2 \left\| [A^{(t)}]^\top QA^\star \beta_i^\star \right\|^2 + 2 \left\| f_i^{(t)} \right\|^2 \leq 4d.$$

Besides, note we have

$$\mathbb{E} \left[ \langle Q\phi_i^\star(x_i^{(j)}), z_i^{(t)} \rangle \langle u, Q\phi_i^\star(x_i^{(j)}) \rangle \langle \beta_i^{(t)}, v \rangle \right] = \langle u, z_i^{(t)} \rangle \langle \beta_i^{(t)}, v \rangle.$$

The proof is then concluded by applying the Bernstein inequality following the same steps as in the final steps of Collins et al. (2021, Proof of Lemma 5). □

We are now ready to control $C_2$.

**Lemma S8.** *Assume **H**1 and let $\delta_d' = cd\sqrt{k}/\sqrt{\lfloor rb \rfloor n}$ for some absolute positive constant $c$. For any $t \in \{0, \ldots, T-1\}$, $\eta > 0$ and whenever $\delta_d' \leq d$, we have with probability at least $1 - e^{-110k} - e^{-110d^2 \log(\lfloor rb \rfloor)}$*

$$C_2 \leq \eta \delta_d' \operatorname{dist}\left(A^{(t)}, QA^\star\right) \left\| \left(R^{(t+1)}\right)^{-1} \right\|_2,$$

*where $C_2$ is defined in (S14), $A^{(t)}$ is defined in (S3) and $R^{(t)}$ comes from the QR factorisation of $\bar{A}^{(t)}$, see step 19 in Algorithm S3.*

*Proof.* Let $t \in \{0, \ldots, T-1\}$ and $\eta > 0$. Then, whenever $\delta_d' \leq d$, we have with probability at least $1 - e^{-110k} - e^{-110d^2 \log(\lfloor rb \rfloor)}$, we have

$$
\begin{aligned}
C_2 &= \frac{\eta}{\lfloor rb \rfloor} \left\| \left( \frac{1}{n}[A_\perp^\star]^\top (Q\mathcal{A}^{(t+1)})^\dagger \mathcal{A}^{(t+1)} \left(Z^{(t+1)}\right) Q - Z^{(t+1)} \right)^\top B_{\mathsf{A}_{t+1}}^{(t+1)} \right\|_2 \left\| \left(R^{(t+1)}\right)^{-1} \right\|_2 \\
&\leq \frac{\eta}{\lfloor rb \rfloor} \left\| \left( \frac{1}{n}(Q\mathcal{A}^{(t+1)})^\dagger \mathcal{A}^{(t+1)} \left(Z^{(t+1)}\right) Q - Z^{(t+1)} \right)^\top B_{\mathsf{A}_{t+1}}^{(t+1)} \right\|_2 \left\| \left(R^{(t+1)}\right)^{-1} \right\|_2 \\
&\leq \eta \delta_d' \operatorname{dist}\left(A^{(t)}, QA^\star\right) \left\| \left(R^{(t+1)}\right)^{-1} \right\|_2,
\end{aligned}
$$

where we used the Cauchy-Schwarz inequality in the second inequality and Lemma S7 for the last one. $\quad\square$

**Control of** $\| \left(R^{(t+1)}\right)^{-1} \|_2$. To finalise our proof, it remains to bound $\| \left(R^{(t+1)}\right)^{-1} \|_2$. The associated result is depicted in the next lemma.

**Lemma S9.** *Define $\bar{\delta}_d = \delta_d + \delta_d'$ where $\delta_d$ and $\delta_d'$ are defined in Lemma S4 and Lemma S5, respectively. Assume **H**1. Then, we have with probability at least $1 - e^{-110k} - e^{-110d^2 \log(\lfloor rb \rfloor)}$,*

$$
\left\| \left(R^{(t+1)}\right)^{-1} \right\|_2 \leq \left( 1 - 4\eta \frac{\bar{\delta}_d}{(1 - \bar{\delta}_d)^2} \bar{\sigma}_{\max,\star}^2 \right)^{-1/2}.
$$

*Proof.* The proof follows from Collins et al. (2021, Proof of Lemma 6). $\quad\square$

## S3 Experimental Details

### S3.1 Data Sets

We provide some details about the datasets we used for our numerical experiments

#### S3.1.1 Toy data sets

The first toy dataset, denoted as *noisy features*, is a 20-class classification problem in which the features for a given class is obtained by sampling a Gaussian distribution of dimension 5, with random mean and Identity covariance matrix. Mean for each class is sampled from a Gaussian distribution with zero mean and diagonal covariance matrix $\sigma^2 \mathbf{I}$ with $\sigma = 0.8$.

For building the training set, we sample 2000 examples for each class and equally share those examples among clients who hold that class. Then, in order to generate some class imbalances on clients, we randomly subsample examples on all clients. For instance, with 100 clients and 2 classes per clients, this results in a problem with a total of about 16k samples with a minimal number of samples of 38 and a maximal one of 400. In order to get different dimensionality, we randomly append on each client dataset some Gaussian random noisy features with dimensionality varying from 1 to 10.

The second toy dataset, denoted as *linear mapping*, is a 20-class classification problem where each class-conditional distribution is Gaussian distribution of dimension 5, with random mean and identity covariance matrix. Mean for each class is sampled from a Gaussian distribution with zero mean 0 and diagonal covariance matrix $\sigma^2 \mathbf{I}$ with $\sigma = 0.5$. matrix. As above, we generate 2000 samples per class, distribute and subsample them across clients in the similar way, leading to a total number of samples of about $15k$. The dimensionality perturbation is modelled by a random (Gaussian) linear transformation that maps the original samples to a space which dimension goes from 3 to 100.

Examples from a given class are equally shared among clients after random permutation of their index. For subsampling samples on a client, we randomly select a percentage of samples to keep ranging from 5% to 100%, and them uniformly randomly select among all classes on the clients the ones we keep.

Table S1: Summary of the Brain-Computer Interfaces dataset we used. We report the number of subjects (#Subj), the number of channels (#Chan), the number of classes (#Classes), the number of trials per class (#Trials class) and the number of features (#features) on the covariance representation has been vectorized.

| Name | #Subj | #Chan | #Classes | #Trials class | # features |
|------|-------|-------|----------|---------------|------------|
| AlexMI | 8 | 16 | 3 | 20 | 136 |
| BNCI2014001 | 9 | 22 | 4 | 144 | 253 |
| BNCI2014002 | 14 | 15 | 2 | 80 | 120 |
| BNCI2014004 | 9 | 3 | 2 | 360 | 6 |
| Weibo2014 | 10 | 60 | 7 | 80 | 1830 |
| Zhou2016 | 4 | 14 | 3 | 160 | 105 |

For these problems, the test sets are obtained in the same way as the training sets but 1000 examples for each class shared across clients.

### S3.1.2 MNIST-USPS

We consider a digit classification problem with the original MNIST and USPS data sets which are respectively of dimension $28 \times 28$ and $16 \times 16$ and we assume that a client hosts either a subset of MNIST or USPS data set. We use the natural train/test split of those datasets and randomly share them accross clients.

### S3.1.3 TextCaps data set

The TextCaps data set (Sidorov et al., 2020) is an Image captioning dataset for which goal is to develop a model able to produce a text that captions the image. The dataset is composed of about 21k images and 110k captions and each image also comes with an object class. For our purpose, we have extracted pair of 14977 images and captions from the following four classes *Bottle*, *Car*, *Food* and *Book*. At each run, those pairs are separated in 80% train and 20% test sets. Examples from the TextCaps datasets are presented in Figure S7. Images and captions are represented by vectors by feeding them respectively to a pre-trained ResNet18 and a pretrained Bert, leading to vectors of size 512 and 768.

Each client holds either the image or the text representation of subset of examples and the associated vectors are randomly pruned of up to 10% coordinates. As such, all clients hold dataset with different dimensionality.

For this dataset, we randomly split the samples for the train and test sets and share them across clients.

### S3.2 Brain-Computer Interfaces data set

The Brain-Computer Interfaces dataset we used are summarized in Table S1. Each dataset description can be obtained from the MOABB library (Jayaram & Barachant, 2018) and at the following URL: http://moabb.neurotechx.com/docs/datasets.html. For each subject, we select the predefined train/test splits or used 75% of the trials for training and the remaining 25% for testing. We used a bandpass prefiltering between 8 and 30 Hz of the EEG signals and extracted a covariance matrix for each trial using all available channels. These covariance matrices are vectorized and used as a feature. The classes that we used for the classification problem are the following ones: ['left hand', 'right hand', 'feet', 'tongue', 'rest'] and a subset of them as available for each dataset. For these datasets, if a subject has more than one session, then we use the last one for generating the test set and the remaining ones for the training set. If we have only a single session, we randomly split the trials into training and test sets.

### S3.3 Models and Learning Parameters

For the toy problems, the *TextCaps data set* and the BCI one, as a local transformation functions we used a fully connected neural network with one input, one hidden layer and one output layers. The number of units in hidden layer has been fixed to 64 and the dimension of latent space as been fixed to 64. ReLU activation

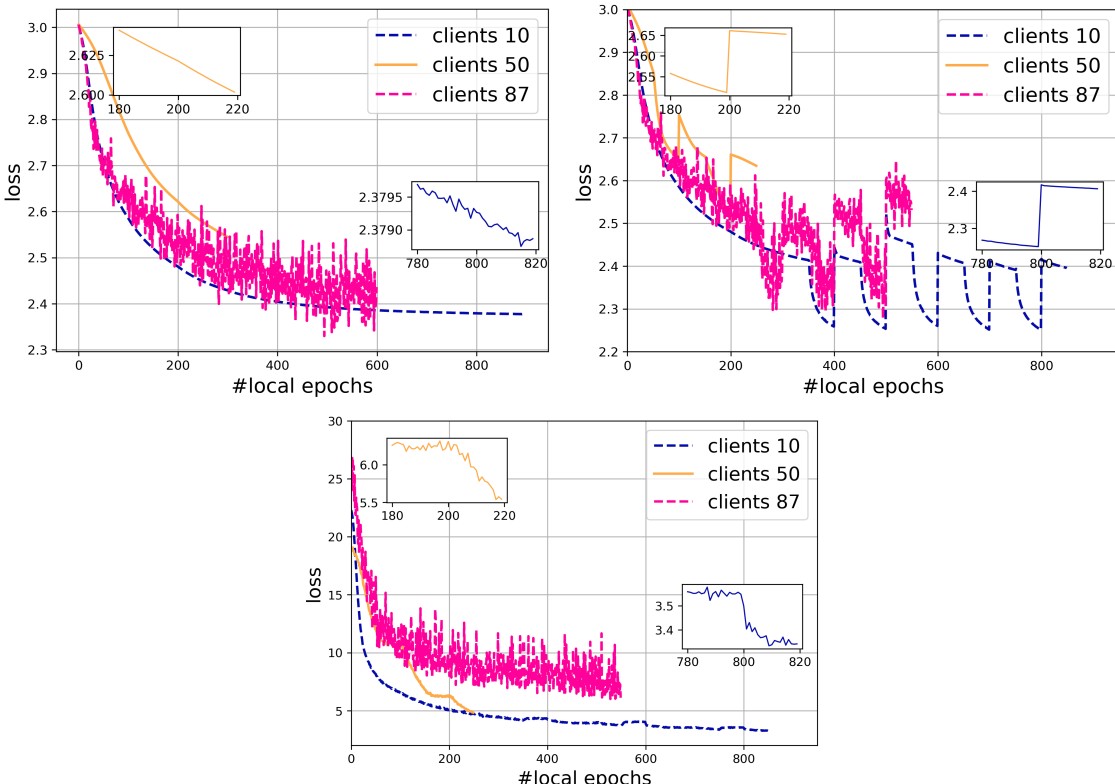

Figure S1: Evolution of the local loss curve of three different clients for three different learning situations. See text for details.

has been applied after the input and hidden layers. For the digits dataset, we used a CNN model with 2 convolutional layers followed by a max-pooling layer and a sigmoid activation function. Once flattened, we have a one fully-connected layer and ReLU activation. The latent dimension is fixed to 64.

For all datasets, as for the local model $g_{\theta_i}$, in order to be consistent with competitors, we first considered a single layer linear model implementing the local classifier as well as a model with one input layer (linear units followed by a LeakyReLU activation funcion) denoting the shared representation layer and an output linear layer.

For training, all methods use Adam with a default learning rate of 0.001 and a batch size of 100. Other hyperparameters have been set as follows. Unless specified, the regularization strength $\lambda_1$ and $\lambda_2$ have been fixed to 0.001. Local sample batch size is set to 100 and the participation rate $r$ to 0.1. For all experiments, we have set the number of communication round $T$ to 50 and the number of local epochs to respectively 10 and 100 for the real-world and toy datasets. For `FLIC`, as in FedRep those local epochs is followed by one epoch for representation learning. We have trained the local embedding functions for 100 local epochs and a batch size of 10 for toy datasets and TextCaps and while of 100 for MNIST-USPS and BCI. Reported accuracies are computed after local training for all clients.

## S3.4   Ablating Loss Curves

In order to gain some understanding on the learning mechanism that involves local and global training respectively due to the local embedding functions, the local classifier and the global representation learning, we propose to look at local loss curves across different clients.

Here, we have considered the *linear mapping* toy dataset as used in the toy problem analysis. However, the learning parameters we have chosen are different from those we have used to produce the results so as to

highlight some specific features. The number of epochs (communication rounds) is set to 100 with a client activation ration of 0.1. Local epochs are shared for either training the local parameters or the global ones (note that in our reference Algorithm 1, the global parameter is updated only once for each client) Those latter are trained starting after the 20-th communication round and in this case, the local epochs are equally shared between local and global parameter updates. Note that because of the randomness in the client selection at each epoch, the total number of local epochs is different from client to client. We have evaluated three learning situations and plotted the loss curves for each client.

- the local embedding functions and the global models are kept fixed, and only the local classifier is trained. Examples of loss curves for 3 clients are presented in the left plot of Figure S1. For this learning situation, there is no shared global parameters that are trained locally. Hence, the loss curve is typical of those obtained by stochastic gradient descent with a smooth transition, at multiple of 100 local epochs, when a given client is retrained after a communication round.

- the local embedding functions are kept fixed, while the classifier and global parameters are updated using half of the local epochs each. This situation is interesting and reported in middle plot in Figure S1. We can see that for some rounds of 100 local epochs, a strong drop in the loss occurs at starting at the 50th local epoch because the global parameters are being updated. Once the local update of a client is finished the global parameter is sent back to the server and all updates of global parameters are averaged by the server. When a client is selected again for local updates, it is served with a new global parameter (hence a new loss value ) which causes the discontinuity in the loss curve at the beggining of each local update.

- all the part (local embedding functions, global parameter and the classifier) of the models are trained. Note at first that the loss value for those curves (bottom plot in Figure S1) is larger than for the two first most left plots as the Wasserstein distance to the anchor distribution is now taken into account and tends to dominate the loss. The loss curves are globally decreasing with larger drops in loss at the beginning of local epochs.

### S3.5   On Importance of Alignment Pre-Training and Updates.

We have analyzed the impact of pre-training the local emebedding functions and their updates during learning for fixed anchor reference distributions . For this, we have kept the same global experimental settings as for the performance comparisons except that we fixed the number of users to 100. We have compared two learning situations : the first one in which the local embedding functions are pre-trained for $K$ epochs and updated during local training and the second in which they are also pre-trained for $K$ epochs and kept fixed during local training. We have reported the classification performance when the number of pre-training epochs $K$ varies from 1 to 200. Results, averaged over 5 runs are shown in Figure S2. At first, we can note that there exists a performance gap between 1 and 200 epochs of pre-training. Nonetheless, this gap is not that large except for the toy added noisy feature dataset. For the three datasets, increasing the number of pre-training epochs up to a certain number tends to increase performance, but overfitting may occur. The latter is mostly reflected in the *toy linear mapping* dataset for which 10 to 50 epochs is sufficient for good pre-training. Examples of how classes evolves during pre-training are illustrated in Figure 5, through *t-sne* projection. We also illustrate cases of how pre-training impact on the test set and may lead to overfitting as shown in Figure S4.

### S3.6   On the Impact of the Participation Rate

We have analyzed the effect of the participation rate of each client into our federated learning approach. Figure S3 reports the accuracies, averaged over 3 runs, of our approach for the toy datasets and the *TextCaps* problem with respect to the partication rate at each round. We can note that the proposed approach is rather robust to the participation rate but may rather suffer from overfitting due to overtraining of local models. On the left plot, performances, measured after the last communication round, for *TextCaps* is stable over participation rate while those performances tend to decrease for the *toy* problems. We associate these decrease to overfitting since when we report (see right plot) the best performance over communication rounds

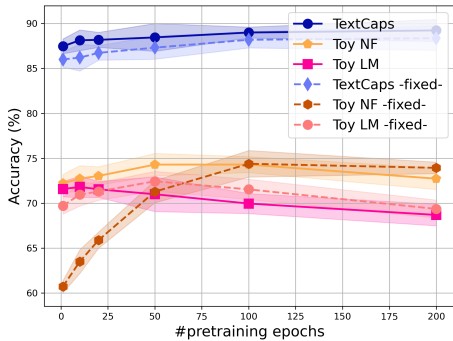

Figure S2: Impact of the number of pre-training epochs for $\phi_i$ on the model accuracy. We compared the case where $\phi_i$ is (plain) updated during local training or (dashed) kept fixed. Results for three different datasets are reported.

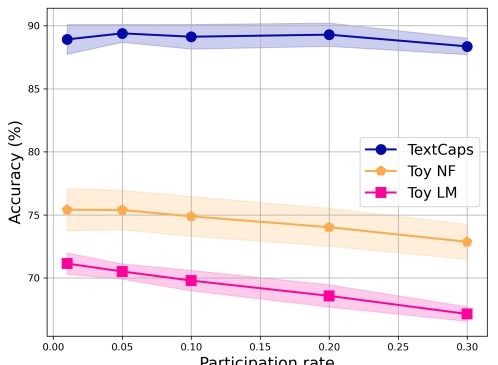 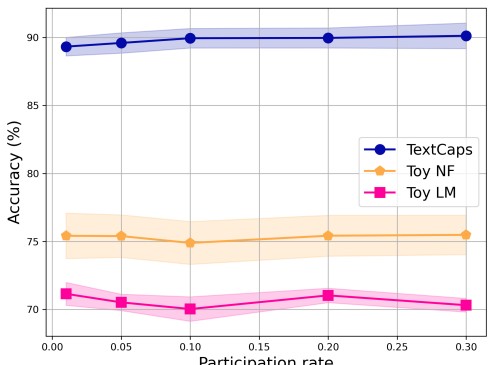

Figure S3: Evolution of the performance of our `FLIC`-Class algorithm with respects to the participation rate of clients, using the same experimental setting as in Figure 4. (left) evaluating performance after last communication rounds, (right) best performance across communication rounds.

(and not the last one), they are stable for all problems. This suggests that number of local epochs may be dependent to the task on each client and the client participation rate.

### S3.7  Comparing `FLIC` with t-sne and `FedRep`

We have claimed several times the need for a proper alignment of the class-conditional distributions when mapping into a common latent subspace. We have illustrated this need in Figure 3 where we have projected the class-conditional distributions of the *toy linear mapping* dataset without alignment and using classical dimensionality reduction algorithm like *t-sne* oracle multidimensional scaling (MDS). Without alignment, mapped class-conditional distributions can get mixed up and learning a accurate classification model becomes hard.

In order to back up our claim with quantitative results, we have compared the performance of `FLIC` with the baseline that consists in projecting data on clients into a common latent subspace (here $\mathbb{R}^{64}$) using *t-sne* and then applying `FedRep` on the projected data. We have followed the same protocol as for the results on toy dataset and reported the results in Figure S5. As expected, the perfomance of this baseline in terms of classification accuracy is lower than the one of `FLIC` and even lower than the one of a local classifier.

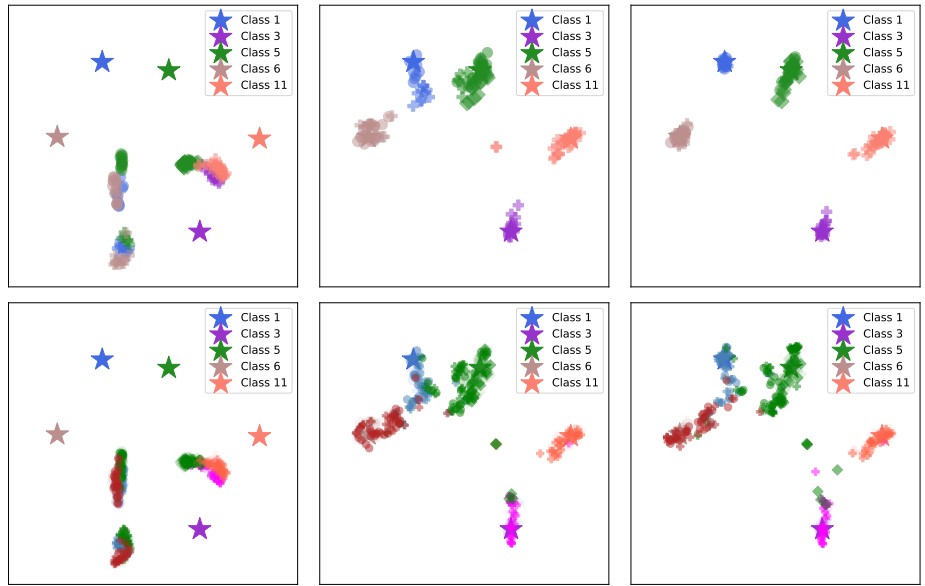

Figure S4: . 2D *t-sne* projection of 5 classes partially shared by 3 clients for the **toy linear mapping** dataset after learning the local embedding functions for (left) 10 epochs, (middle) 50 epochs, (right) 100 epochs. Original dimensions on clients vary from 5 to 50. Top row shows the projection the training set while bottom row plots show both training and test set. Star ⋆ markers represent the projection of the mean of each class-conditional. The three different marker styles represent the different clients. Classes are denoted by colors and similar tones of color distinguish train and test sets. We see that each class from the training set from each client converges towards the mean of its anchor distribution, represented by the star marker. Interestingly, we also remark that unless convergence is reached, empirical class-conditional distributions on each clients are not equal making necessary the learning of a joint representation. From the bottom plots, we can understand that distribution alignment impacts mostly the training set but this alignment does not always generalize properly to the test sets.

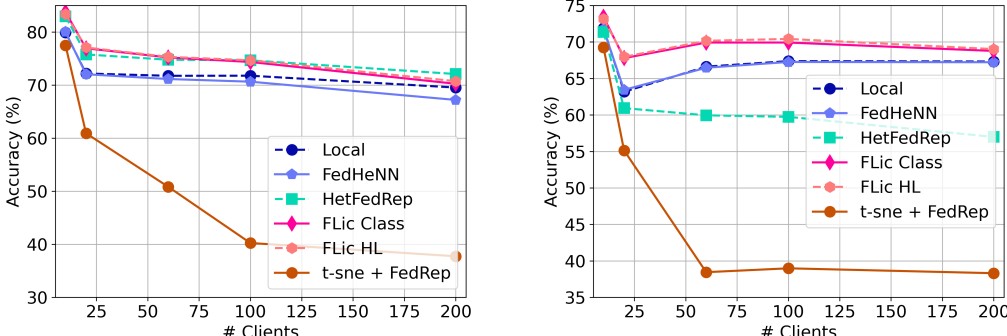

Figure S5: Comparing the baseline *t-sne* projection + `FedRep` to `FLIC` and other baselines We report the classification accuracy for the (left) *toy noisy features* dataset and (right) *toy linear mapping* dataset.

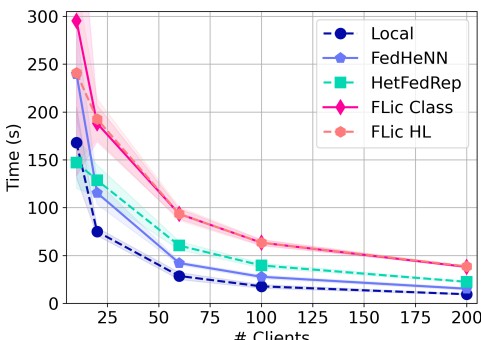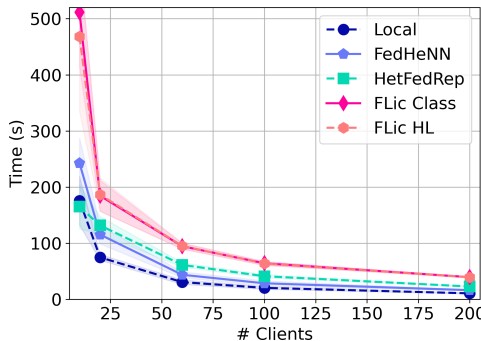

Figure S6: Comparing the computational cost for training all the models over 50 epochs on the toy dataset (left) *toy noisy features* dataset and (right) *toy linear mapping* dataset.

### S3.8 On the computational cost of `FLIC`

The approach we propose has computational overhead compared to the classical federated learning approach and compared to the baselines that are able to handle heterogeneous feature spaces. For instance, in addition to the local version of the model on each client, we need to update the local embedding function and the anchor distributions parameters.

In order to properly quantify the computational cost of `FLIC`, we report in Figure S6 the running time needed for training each model over 50 epochs and for increasing number of clients, for the toy dataset problems. Remind that as the number of clents increase, the number of samples on each client decreases. We can see that roughly, the computational cost of `FLIC` is about two times higher than for other competitors, across the range of clients and for the two learning problems.

### S3.9 Experiments on `FEMNIST`

We have also evaluated our approach on the `FEMNIST` dataset. It is part of the LEAF (LEAF: A Benchmark for Federated Settings) project (Caldas et al., 2018), which aims to provide a benchmarking framework for learning in federated settings. It is a 28 by 28 pixel image dataset that is built by partitioning the data in the Extended MNIST dataset based on the writer of the digit/character. It contains 62 different classes, including 10 digits, 26 lowercase letters, and 26 uppercase letters. As in Collins et al. (2021), we have considered a subset of 50 and 150 clients (one writer, one client) and limited ourselves to the 10 digits samples available for each writer. Samples in `FEMNIST` are all of the same dimensionality so our goal here is to compare our approach to some baselines such as `FedRep` for a well-known federated learning problem. For training, we used the same parameters as for the other datasets except that the number of epochs has been set to 1000, as the learning problem is more complex and needs more epochs to converge.

Results, averaged over 3 runs, are reported in Table S2. We first note that the performance of `FedRep` is in line with the ones reported in Collins et al. (2021). Then, we remark that our approach outperforms the competitors on this dataset by a large margin with a performance gain of about 5% over the best competitor. This is a strong result that shows the potential of our approach on real-world datasets even when the feature spaces are homogeneous. We can also highlight the impact of pre-training the local embedding functions when comparing the performance of `FLIC` and `HetFedRep` (the latter is a variant of `FLIC` where the local embedding functions are not pre-trained). Here, pre-training the local embedding functions play the role of a domain adaptation mechanism that allows to better align the class-conditional distributions of each client with a common shared representation.

Table S2: Performance comparison on the `FEMNIST` dataset. We report the classification accuracy for the `FEMNIST` dataset for 50 and 150 clients.

| METHOD | 50 CLIENTS | 150 CLIENTS |
|---|---|---|
| LOCAL | $77.29 \pm 0.6$ | $79.59 \pm 0.7$ |
| FEDHENN | $71.01 \pm 0.8$ | $74.77 \pm 0.2$ |
| HETFEDREP | $52.90 \pm 2.0$ | $57.78 \pm 2.3$ |
| FEDREP | $70.29 \pm 3.0$ | $74.55 \pm 1.3$ |
| FLIC-CLASS | $82.25 \pm 0.4$ | $\mathbf{84.81 \pm 0.4}$ |
| FLIC-HL | $\mathbf{82.37 \pm 0.6}$ | $84.30 \pm 0.4$ |

A pan sits on a hob with a lid bearing the logo Hamilton Beach Stay n Go

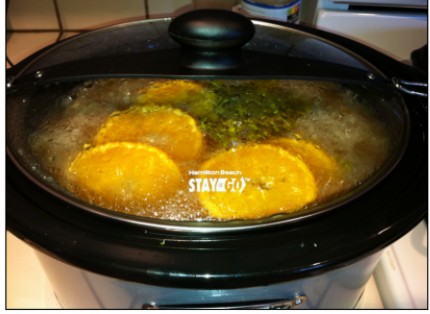

Deciding on whether to drink spring water or a 7UP.

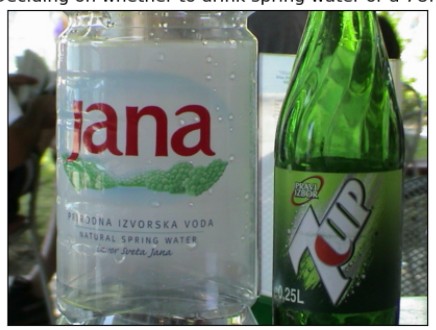

The bookshelf consists of about 20 different types of books.

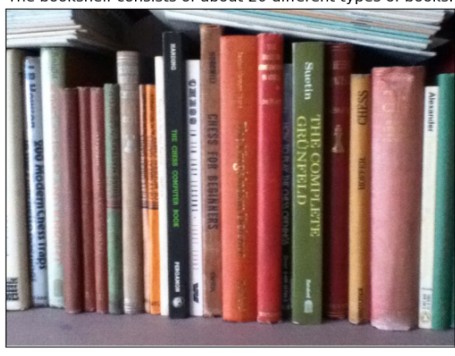

The yellow convertible is on display somewhere in California.

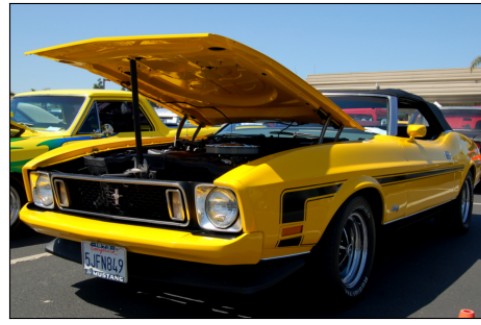

Figure S7: Examples of some TextCaps pairs of image/caption from the 4 classes we considered of (top-left) Food, (top-right) Bottle, (bottom-left) Book (bottom-right) Car. We can see how difficult some examples can be, especially from the caption point of view since few hint about the class is provided by the text.

