# OpenReview forum: "Personalised Federated Learning On Heterogeneous Feature Spaces"
_TMLR — Accepted by TMLR_

### Review · Reviewer_eR4X · 2024-01-21

**Summary Of Contributions:**

The paper formulizes personalized Federated Learning on heterogeneous client’s feature spaces and proposes a framework FLIC. FLIC learns a function locally at each client to map the input feature space to a common embedding space. To preserve semantical information from the original data distribution, the locally learned functions at each client are aligned with some learnable latent anchor distributions that are shared across all clients. A  personalized Federated Learning algorithm is then employed on the common embedding space to learn the desired task. The paper shows that for a simpler regression framework where the anchor distribution is known beforehand and not learned under the FL paradigm, FLIC can recover the true latent subspace underlying the FL problem. The effectiveness of FLIC is shown through experiments on toy and real-world problems including heterogeneous BCI datasets.

**Audience:**

Yes

**Claims And Evidence:**

Yes

**Requested Changes:**

See Weaknesses/questions

**Strengths And Weaknesses:**

Strengths:
1. The paper proposes a novel framework for federated learning on heterogeneous feature spaces via the mapping of input features into a common embedding space
2. Theoretical analysis for a simple regression framework is presented.
3. Experimental results are presented for a toy example and real-world datasets such as Digits, TextCaps, and BCI datasets.
4. The paper discusses the limitations and privacy impacts of the proposed framework.
5. The appendix includes interesting ablation studies.

Weaknesses/questions:
1. The paper was hard to follow and can use some improvement in writing.
2. It is not clear how the anchor distributions are initialized. In particular, what is $v_{1:C}^0$ in Algorithm S2? Is it equal to $\hat{m}_{1:C}^0$?
3. FLIC has memory and communication overheads stemming from $v_{i, 1:C}, L_{i, 1:C}$.
4. FLIC is computationally heavy at the client end as it needs to compute gradients with respect to multiple parameters and loss functions.
5. It would be helpful to present some quantitative results comparing the overheads of FLIC to the other baselines.
6. The Averaging anchor distribution paragraph on page 7 mentions "We provide algorithmic details regarding steps
14 and 20 in Algorithm 1". Algorithm 1 only has 18 steps and that paragraph doesn't seem to be consistent with the algorithm.
7. Section S3.5 and the caption for Figure S2 are not clear.  It mentions "We have considered two learning situations: one in which they are
updated during local training (as usual) and another one in they are kept fixed all along the training". Is the transformation $\phi$ pre-trained in these two cases? The discussion on the impact of pre-training the embedding space is interesting and needs to be more clearly written.

---

> ### Author Response · Authors · 2024-03-02
> **reply to review**
>
> We would like to thank the reviewer for the insightful comments and suggestions.  We have addressed them as detailed below.
>
> ### Comments
>
>
> > It is not clear how the anchor distributions are initialized.
>
> We initialize the mean vectors $v_c$ of the anchor distributions by randomly sampling a Gaussian distribution with a very large variance. By doing so, we ensure a good separability of the anchor distributions in the latent space. The covariance matrices are initialized as identity matrices.
>
> We have added this clarification to the pretraining paragraph in the Section 4.
>
> >  In particular, what is $v_{1:C}^0$ in Algorithm S2? Is it equal to $\hat m_{1:C}^0$
>
> $v_{1:C}$ are the means of the anchor distributions and $\hat m_{1:C}$ are the means of the mapped distributions. As stated in Section 3 Design choices, we have modeled both the  anchor distributions and the mapped distributions as Gaussian distributions.
>
> We have added this  clarification in the algorithm description.
>
> .
> > FLIC is computationally heavy at the client end as it needs to compute gradients with respect to multiple parameters and loss functions. It would be helpful to present some quantitative results comparing the overheads of FLIC to the other baselines.
>
> We have added a paragraph to discuss the computational overhead of FLIC compared to the other baselines. In short, we have evaluated the running time of the different methods for running the $50$ epochs on the toy datasets and found that FLIC is computationally heavier than the other methods (about a factor of two).
>
> We have added this paragraph and a supporting figure in the Appendix Section S.3.8.
>
> > The Averaging anchor distribution paragraph on page 7 mentions "We provide algorithmic details regarding steps 14 and 20 in Algorithm 1". Algorithm 1 only has 18 steps and that paragraph doesn't seem to be consistent with the algorithm.
>
> Thanks for pointing this out. We have corrected this mistake and clarified the paragraph.
>
> > Section S3.5 and the caption for Figure S2 are not clear. It mentions "We have considered two learning situations: one in which they are updated during local training (as usual) and another one in they are kept fixed all along the training". Is the transformation
>  pre-trained in these two cases? The discussion on the impact of pre-training the embedding space is interesting and needs to be more clearly written
>
> We have rewritten the paragraph and the caption to clarify the two learning situations. In the first case, the transformation is pre-trained and updated during local training. In the second case, the transformation is pre-trained and kept fixed all along the training.
>
> We corrected the paragraph accordingly.

---

### Review · Reviewer_beo7 · 2024-02-15

**Summary Of Contributions:**

The paper presents a method for applying federated learning (FL) to heterogeneous datasets, which may have different schema or feature spaces. In order to enable FL, which typically assumes all client data has been drawn from the same feature space, across clients with distinct feature spaces, the authors propose having each client learn a local embedding function that maps client data to a common subspace. This framework "FLIC" is compatible with existing FL algorithms such as FedRep and FedAvg with fine-tuning. The authors include some empirical results on a range of datasets - from toy to brain-computer interface data - and some theoretical analysis of the proposed approach.

Primary contributions:
* First FL framework for operating on client data with heterogeneous feature spaces.
* First empirical results for using FL on brain computer interface data.
* Distribution alignment algorithm for learning local embeddings functions to map data to a common subspace.

**Audience:**

Yes

**Broader Impact Concerns:**

Learning on heterogeneous data formats has the potential to enable learning across data collected by different institutions without standardizing the format a priori. Making the claim of expanded privacy-aware FL scope, however, is misleading without adding a privacy analysis to strengthen "Remark 3" and demonstrating that there is no further privacy leakage than what is given by FedAvg. Stating how the proposed approach might be compatible with privacy techniques such as differential privacy or secure aggregation would strengthen the work.

**Claims And Evidence:**

No

**Requested Changes:**

Critical changes:
* Additional experimental results would strengthen the work.
    * Include baseline performance of FLIC on federated datasets commonly used in the literature, where the partitions are based on user and all data has the same format. This would indicate how well the method works on data that is non-iid but not completely disjoint, and offer a comparison point to common FL approaches.
    * Demonstrate FedAvg + FT with learned feature embeddings for comparison (simpler baseline for personalized FL).
    * Compare FLIC to t-sne and multi-dimensional scaling in the experimental results. Depicting clusters visually using these methods on a toy set does not demonstrate the effect of these methods in practice.

* Additional description of figures and discussion of what the results show is necessary.
    * Figure 2: It is not clear how the two plots included in the figure relate. Either the relation should be discussed or the plots should be separated. What should be understood from the leftmost plot? There is no discussion and no metrics to indicate whether the projected features are far from the ground truth. The caption of the rightmost plot includes numbers that are not matching what is shown and requires further explanation.
    * Figure 4: What is the effect of increasing the number of clients on a toy problem? Is this a more or less challenging problem? Include analysis of the results and interpretation of how or why each method differs.

* Section 4 claims that the presented algorithm avoids the client drift phenomenon. It is unclear how this is possible, as the federated algorithm presented has clients train locally for some number of steps between rounds of parameter averaging. The communication cost of using this approach is strictly higher than other federated algorithms without it, such as FedAvg or FedRep, due to the additional parameters that are needed to define the learned anchor measures. Claiming FLIC is communication-efficient is misleading and incorrect.

Minor changes:
* Section 4 “Averaging Anchor Distributions” references incorrect steps in Algorithm 1.
* The experimental section could use further clarification. Is the reported accuracy on a held out portion of each client’s data?
* It is not clear what should be expected when plotting performance with respect to the number of clients.
* Table 4 accuracies should include confidence intervals over the 3 runs.
* Simplifying the notation through the manuscript and algorithms would help with the readability. As this approach is compatible with different model definitions it would be useful to use a simple base model \theta in the algorithms and throughout the text, rather than splitting the model into \alpha and \beta components, since that is not the contribution of this work. The key change this work presents is in adding a local embedding layer, which could be expressed much more simply.
* Typo “jsut” on page 10.
* A limitation that should be noted is that this is a stateful algorithm applicable to the cross-silo setting.

**Strengths And Weaknesses:**

Strengths:
* Defines a strategy for learning client-specific embeddings across heterogeneous feature spaces in a federated way.
* The need for such an approach is well-motivated, especially in the case of cross-institution FL where dataset formats may not be standardized a priori.
* This learned feature representation method is broadly applicable to existing FL algorithms.

Weaknesses:
* There is a need for specifying in advance the latent dimensionality and having some prior knowledge of the latent distribution.
* The theoretical analysis is for a simplified setting in which the anchor distribution is known beforehand, which may not be the case in practice.
* Lacks comparison with alternative ways of embedding distinct data in a common subspace.
* The empirical results are quite simple. The chosen datasets and settings do not have prior work on them to have benchmarks and points of reference that could be used to put the numbers in context in terms of expected performance. The difference in performance across methods are quite minimal with the top accuracy falling within the confidence intervals of other approaches (e.g. Table 3).
* The paper anchors the contribution and discussion on extending FedRep to the heterogeneous feature space setting, making it difficult to understand for anyone unfamiliar with the prior work, even though the contributions are more broadly applicable.

---

> ### Author Response · Authors · 2024-03-02
> **reply to review**
>
> We would like to thank the reviewer for the insightful comments and suggestions.  We have addressed them as detailed below.
>
>
> ### Comments on Required Changes
>
> > The difference in performance across methods are quite minimal with the top accuracy falling within the confidence intervals of other approaches (e.g. Table 3).
>
> While we can agree that performance differences are minimal, compared to the best competitors, we believe that the performance of FLIC is consistently better than the other methods. For instance, FedHENN achieves a bit worst performance than FLIC on the digit dataset (from 0.5\% to 4%) and on TextCaps, the gap is consistently about 5\% of performance.
>
> Also note that our approach can also be seen as a way to (slightly) improve the performance of classical FL algorithm, for problem with same feature spaces.
>
>
> > Include baseline performance of FLIC on federated datasets commonly used in the literature, where the partitions are based on user and all data has the same format. This would indicate how well the method works on data that is non-iid but not completely disjoint, and offer a comparison point to common FL approaches
>
> We would be happy to include results on other datasets with user-based partitions, but it seems to us that most paper uses CIFAR datasets. So we would be grateful if the reviewer could provide us with a reference to a federated dataset with such characteristics.
>
>
>
> > Other Baselines: FedAvg + FT with learned feature embeddings for comparison (simpler baseline for personalized FL).
>
> Due to lack of time, we were not able to implement this baseline. We will include it in the final version of the paper. However, as our goal is to show the effectiveness of FLIC in learning local embeddings, we believe that the comparison using FedAvg + FT instead of FedRep would not bring extra information compared to the baselines we already implemented.
>
>
> > Compare FLIC to t-sne and multi-dimensional scaling in the experimental results. Depicting clusters visually using these methods on a toy set does not demonstrate the effect of these methods in practice.
>
> The comparison with t-sne and multi-dimensional scaling is not fully relevant in our case. Indeed, these methods can not be applied with out-of-samples and so would require to have access to the full dataset (train and test set) to be relevant. We believe that this is a strong assumption.
>
> A possible way to apply these methods is to distill the t-sne or mds projection into a neural network. In this case, we would need to learn the projection and then train a neural network on the projected data. This approach would be somehow similar to ours but would require a two-step learning process.
>
> However, in order to support our rationale on the need for alignment with anchor distributions, we have run experiments on the toy problem datasets and compared the performance of FLIC with t-sne and mds. The results are presented in the Appendix section S.3.1.2.  We show there that as expected, t-sne and mds are not able to capture the class information of the data and lead to poor classification performance.
>
>
> >  2 Figure 2: It is not clear how the two plots included in the figure relate
>
> We agree that this figure is not clear, especially the left plot and we have removed it from the paper. We have rewritten the caption in order to (hopefully) clarify the figure.
>
>
> > Figure 4 what is the effect of increasing the number of clients on a toy problem? Is this a more or less challenging problem
>
> The effect of increasing the number of clients on the toy problems is to decrease the number of samples per client. This makes the learning problem harder. We have added this explanation in the paper and have provided a detailed explanation in Section 6 "Results on toy datasets".
>
> > Section 4 claims that the presented algorithm avoids the client drift phenomenon...
>
> We agree that section 4 includes some statements that are not supported by the experiments. We have removed these statements from the paper.

---

> > ### Comment · Reviewer_beo7 · 2024-03-19
> >
> > Thank you to the authors for making revisions to address many of the comments made.
> >
> > One outstanding revision that should be addressed in the final version of the paper: Algorithm 1, step 20 is referred to in the text but the algorithm only has 19 lines.
> >
> > To answer the authors' question about what federated datasets with natural partitioning might exist in the literature: EMNIST is a federated version of MNIST where each client corresponds to the author who wrote a collection of handwritten characters. See: https://arxiv.org/abs/1812.01097, https://github.com/TalwalkarLab/leaf/tree/master/data/femnist, https://www.tensorflow.org/federated/api_docs/python/tff/simulation/datasets/emnist/load_data. EMNIST is a standard FL baseline that could be used to measure performance of FLIC in a setting where performance is well understood, partitions are based on user and all data has the same format. This would indicate how well the method works on data that is non-iid but not completely disjoint, and offer a comparison point to common FL approaches.

---

> > > ### Author Response · Authors · 2024-03-29
> > > **minor updates**
> > >
> > > Thanks to the reviewer for the feedbacks. We apologize for missing the reference to step 20 in the paper. We have corrected this point in the revised version of the paper.
> > >
> > >  We have also added a new experiment on FEMNIST to compare the performance of our method with our baselines on this new dataset. The results are presented in the appendix S3.9 and clearly show the superiority of our method over the baselines even in a situation where feature spaces are homogeneous.

---

> ### Author Response · Authors · 2024-03-02
> **reply to review**
>
> ### Comments on minor changes
>
> > Section 4 “Averaging Anchor Distributions” references incorrect steps in Algorithm 1.
>
> We have corrected those errors.
>
> > The experimental section could use further clarification. Is the reported accuracy on a held out portion of each client’s data?
>
> In some cases, we have used the natural train/test split of the dataset. In other cases, we have used a random split. We have clarified this point in the appendix section S.3.1.1.
>
> > It is not clear what should be expected when plotting performance with respect to the number of clients.
>
> For Figure 2, as the number of clients increases, the number of samples per client decreases. This makes the learning problem harder. We have added this explanation in the paper and have provided a detailed explanation in Section 6.
>
> > Table 4 accuracies should include confidence intervals over the 3 runs.
>
> we have added the standard deviation to our results, which was missing due to space constraints.
>
> > Simplifying the notation through the manuscript and algorithms would help with the readability. As this approach is compatible with different model definitions it would be useful to use a simple base model \theta in the algorithms and throughout the text, rather than splitting the model into \alpha and \beta components, since that is not the contribution of this work. The key change this work presents is in adding a local embedding layer, which could be expressed much more simply.
>
> We agree with the reviewer that  our contribution is the local embedding layer and that the key algorithmic change is the learning end-to-end of those local embeddings layers and the federated model.  Hence, for the definition of the objective function and the description of the algorithm, as suggested by the reviewer, we have kept the notation $\theta$ for the model parameters.
>
> However since our implementation follows FedRep, for a sake of clarity, we have kept the notation $\alpha$ and $\beta$ in the algorithm description. We have added a note in the algorithm description to clarify this point.
>
>
>
> We have corrected all other typos and minor issues raised by the reviewer.

---

### Review · Reviewer_5DVB · 2024-02-19

**Summary Of Contributions:**

The authors propose a federated learning algorithm to handle cases where the datasets and features are  heterogeneous across clients. The key ingredients of the proposed FLIC framework are the following:
* Map client data to a common representation space with learnable mapping $\phi_i$ for client $i$.
* In the common representation space, encourage examples from class $c$ from all clients to be close to an anchor Gaussian distribution $\mu_c$.
* On top of these representations, learn a classifier parameterized by global parameters $\alpha$ and local parameters $\beta_i$ for clients $i \in [b]$.

The authors apply their methods to
* Two toy datasets with Gaussian data
* MNIST and USPS digit classification task
* An image, caption to class task
* A Brain Computer Interface data task.

**Audience:**

Yes

**Broader Impact Concerns:**

None.

**Claims And Evidence:**

Yes

**Requested Changes:**

Details for experiments should be fully specified. For example, in the toy dataset problems, the following details should be added.
* How are the means of the class-conditional distributions generated? How separated are the means compared to the noise?
* How are examples subsampled in each client?
* How is the random matrix formed?

For the plot in Figure 3 (left), more details are required for reproducibility. For example, the Gaussian distributions used to generate data for the 10 classification problems.

In Figure 4 (left), why is the accuracy going down as we increase the number of clients? In Figure 4 (right), why does it look very different from 4 (left)? Why is HetFedRep so poor the right plot, but good on the left plot?

In Theorem 1:
* The result is on regression, whereas the focus of FLIC is classification.
* Are you assuming that $k_i \ge k$ for $i \in [b]$? If so, consider making it explicit.
* Suppose $k_i = k$ for simplicity. Assume $\Sigma = P D P^T$. May be I am missing something, but should the mapping from $N(0, \Sigma)$ to $N(0, I)$ be
$f(x) = D^{-1/2} P^T x$ instead of $f(x) = D^{-1/2} x$?
Note that, for $y=Bx$,
$$E[yy'] = E[Bxx'B'] = B \Sigma B' = D^{-1/2} P' PDP' (D^{-1/2}P')' = I$$
You can apply another orthonormal transformation and still the covariance would be identity. So, there is nothing special about the diagonal matrices with $\pm 1$, I think.


**Following comments are minor in nature**

Why don't we explicitly enforce separation between means $\mu_c$ of different classes? Is it automatically done by the classification loss minimization?

What does FLIC stand for?

It should be mentioned in the main text how $\phi_i$ can be parametrized.


Typo: In page 7, right below Algorithm 1, the description of the algorithm, the step numbers seems to be wrong.

Typo: Correct the typo "FedRed".

What is principal angle distance between two orthonormal matrices A, B?

**Strengths And Weaknesses:**

The proposed FLIC framework is interesting -- in particular, the alignment of representations of same-class examples with a latent anchor distribution. In numerical experiments, the proposed methods FLIC-HL and FLIC-Class perform well on these problems.

The FLIC methods have only marginal gains over competing methods. The theoretical result (Theorem 1) appears unsatisfactory.

---

> ### Author Response · Authors · 2024-03-02
> **reply to review**
>
> We would like to thank the reviewer for the insightful comments and suggestions.  We have addressed them as detailed below.
>
>
>
> > * How are the means of the class-conditional distributions generated? How separated are the means compared to the noise?
> > * How are examples subsampled in each client?
> > * How is the random matrix formed?
>
> we initialize the means of these anchor distributions by sampling a Gaussian distribution with a very large variance ensuring a large separation between the means of the different anchor distributions.
>
> Regarding the other points, we have provided more details that have been added in the Appendix section S.3.1.1
>
> > For the plot in Figure 3 (left), more details are required for reproducibility. For example, the Gaussian distributions used to generate data for the 10 classification problems.
>
> For a sake of consistency, we have changed the dataset used in this figure. We now use the "added noisy feature" toy problem described in the Appendix section S.3.1.1. and also used for reporting performance in Figure 4. We thus have a new figure but the same conclusions that without a proper alignment to anchor distributions,  class-conditions  from different classes and clients may overlap yielding a poor classification performance. For supporting the latter, we also added an experiment in the Appendix Section 3.7.
>
> > In Figure 4 (left), why is the accuracy going down as we increase the number of clients? In Figure 4 (right), why does it look very different from 4 (left)? Why is HetFedRep so poor the right plot, but good on the left plot?
>
> We believe that the accuracy is going down as the total number of samples is fixed for the experiments and thus when number of clients increases, the number of samples per client decreases making the learning problem harder.
>
> One rationale we can provide for the poor performance of HetFedRep for the "linear mapping" toy problem is the nature of the problem.
> For the "added noisy features" problem, each local feature transformation in HetFedRep aims at finding the true features of the problem, while for the linear mapping problem, the local feature transformation seeks at ``inverting'' the linear mapping which is presumably a harder problem.

---

> ### Author Response · Authors · 2024-03-02
> **reply to review**
>
> > Theoretical result is on regression, whereas the focus of FLIC is classification.
>
> We acknowledge that our theoretical analysis provides only a partial result which establishes the convergence of the algorithm. However, we firmly believe that it offers insightful perspectives, and it can be important to recognize the significance of those partial result, even if they do not provide a full understanding of the behavior of the method.
>
>
> > $k_i \geq k$
>
> Indeed, in the proof we assume that $k_i \geq k$. We have made clear this assumption in the hypothesis H1 of the theorem.
>
> > Suppose for simplicity that...
>
> Based on theorem 4.1 in Delon 2022, we agree with reviewer that there is an error here on the formulation of the mapping. We have corrected this mistake in the paper (see Appendix S2.1)
>
>
>
> ### Comments on minor changes
>
> > Why don't we explicitly enforce separation between means
>  of different classes? Is it automatically done by the classification loss minimization?
>
> In the learning problem defined in Equation 3, we could have enforced directed separation between the means of the different anchor distributions. Here instead, we have chosen (in the second and third terms) to optimize those means with respect to the classification loss and with respect to the mapped distributions of the classes. Also note that in practice, we initialize the means of these anchor distributions by sampling a Gaussian distribution with a very large variance ensuring a large separation between the means of the different anchor distributions.
>
> We have added this explanation in the paper Section 4.
>
> > What does FLIC stand for?
>
> FLIC is an acronym that we obtain by cherry-picking letters from "Federated LearnIng from heterogeneous feature spaCes". It is easy to pronounce and remember and stands for "cop" in french language.
>
>
>
> > It should be mentioned in the main text how $\phi_i$ can be parametrized.
>
> We have added this explanation in the paper Section 3 below remark 2
>
>
> > What is principal angle distance between two orthonormal matrices A, B?
>
> The principal angle distance is a standard metric for comparing two subspaces.  We have its formal definition in the first page of the appendix.

---

### Author Response · Authors · 2024-03-02
**Reply to review**

We would like to thank the reviewers for their insightful comments and suggestions. We have carefully read and considered all the comments and have made the changes to the paper which have been highlighted in blue.  We have also addressed the comments and suggestions as detailed below.

---

### Decision · Action_Editor_34BZ · 2024-04-01

**Recommendation:** Accept as is

**Comment:**

The authors propose FLIC, a framework that enables personalized FL in scenarios where different clients may have different data representations (heterogeneous feature spaces). This involves clients learning functions to map their data to a common feature space, and the embedding functions being optimized such that the learned feature space represents similar data/same classes across clients similarly. The authors provide simplified theoretical analysis showing that their approach recovers the true latent subspace. The experiment shows some gains, even if marginal, over other embedding methods. The authors incorporated presentation improvements (clarified concepts, algorithm steps, and provided missing details on the experimental setup, removed some statements not supported by the experiments), and ran additional experiments comparing to other baselines as suggested by the reviewers, further improving the paper.

**Audience:**

The problem of personalized FL on heterogeneous client feature spaces is important especially in real-world scenarios like medical domains where data formats may not be standardized. The paper introduces a new framework specifically designed for this problem, offering a different angle than existing methods. The successful application to Brain-Computer Interface datasets hints at the potential impact in domains with similar data variability challenges, making it relevant to researchers across different fields.

Overall, the work presents a valuable contribution to the field of federated learning, which is of interest to the TMLR audience.

**Claims And Evidence:**

The claims reported in the submission are accurate (and the ones lacking support have been removed), and come with theoretical support in simplified scenarios, and a reasonable amount of empirical evidence on toy and real datasets (including a pioneering application to Brain-Computer Interfaces).

The theory is built for a simple regression task, and requires certain assumptions which limits its direct applicability. More thorough comparisons with other embedding strategies would make the claims about FLIC's approach stronger.